# Speedup Patch: Learning a Plug-and-Play Policy to Accelerate Embodied Manipulation

**Zhichao Wu** [1 2 *]  **Junyin Ye** [1 2 *]  **Zhilong Zhang** [1 2 *]  **Yihao Sun** [3]  **Haoxin Lin** [1 2]
**Jiaheng Luo** [1 2]  **Haoxiang Ren** [1 2]  **Lei Yuan** [1 2]  **Yang Yu** [1 2 †]

## Abstract

While current embodied policies exhibit remarkable manipulation skills, their execution remains unsatisfactorily slow as they inherit the tardy pacing of human demonstrations. Existing acceleration methods typically require policy retraining or costly online interactions, limiting their scalability for large-scale foundation models. In this paper, we propose **S**peed**u**p **P**atch (**SuP**), a lightweight, policy-agnostic framework that enables **plug-and-play acceleration** using solely offline data. SuP introduces an external scheduler that adaptively downsamples action chunks provided by embodied policies to eliminate redundancies. Specifically, we formalize the optimization of our scheduler as a Constrained Markov Decision Process (CMDP) aimed at maximizing efficiency without compromising task performance. Since direct success evaluation is infeasible in offline settings, SuP introduces **world model based state deviation** as a surrogate metric to enforce safety constraints. By leveraging a learned world model as a virtual evaluator to predict counterfactual trajectories, the scheduler can be optimized via offline reinforcement learning. Empirical results on simulation benchmarks (Libero, Bigym) and real-world tasks validate that SuP achieves an overall $1.8\times$ execution speedup for diverse policies while maintaining their original success rates.

## 1. Introduction

Recent advances in embodied intelligence have demonstrated the potential of generalizable robotic manipulation. By learning from large-scale demonstration datasets, embodied policies can preform fine-grained manipulation tasks, comprehend natural language instructions, and draw up clear task plans (Zhao et al., 2024; Black et al., 2024; 2025a; Zhang et al., 2026a). However, despite these capabilities, current embodied policies often suffer from execution inefficiency, completing tasks at a slow pace that hinders real-world deployment (Park et al., 2024; Guo et al., 2025).

This execution inefficiency stems primarily from the redundant nature of human teleoperation data, characterized by the slow movement patterns of human demonstrators (Guo et al., 2025). Consequently, trained policies tend to output excessively dense action sequences, resulting in long task execution times. Existing methods for execution acceleration often come with expensive costs. They necessitate intricate data curation and policy retraining (Guo et al., 2025; Kim et al., 2025), introduce entirely new action prediction mechanisms (Arachchige et al., 2025), or rely on expensive online interactions to maintain task success (Yuan et al., 2025; Nam & Hwang, 2025). Such requirements lead to substantial training overhead and limited scalability, especially for large Vision-Language-Action (VLA) models where fine-tuning is computationally expensive. This motivates our core inquiry: *Can we achieve plug-and-play acceleration for diverse embodied policies using solely offline data, without retraining the original policies?*

To bridge the gap, we propose **S**peed**u**p **P**atch (SuP), a lightweight, policy-agnostic framework that learns an external scheduler policy from offline demonstration data alone. The scheduler acts as a "patch" that adaptively downsamples the action chunks provided by the embodied policy to eliminate redundancies. We formalize the optimization of our scheduler as a Constrained Markov Decision Process (CMDP) (Gu et al., 2024; Zhao et al., 2023b), aimed at maximizing execution efficiency without compromising task performance.

However, solving this CMDP in offline setting is non-trivial,

---

*Equal contribution  [1]National Key Laboratory for Novel Software Technology, Nanjing University, China [2]School of Artificial Intelligence, Nanjing University, China [3]Mila-Quebec AI Institute & Université de Montréal, Canada. Correspondence to: Yang Yu <yuy@nju.edu.cn>.

*Proceedings of the $43^{rd}$ International Conference on Machine Learning*, Seoul, South Korea. PMLR 306, 2026. Copyright 2026 by the author(s).

as evaluating task success typically necessitates costly online interaction. To avoid this, we propose *world model based state deviation*—defined as the discrepancy between the end-effector (EEF) trajectories of the original and accelerated actions—as a surrogate safety constraint to bound the acceleration rate. Specifically, we first train a world model to capture the environment dynamics from existing demonstrations. We then leverage this model as a virtual evaluator to generate counterfactual trajectories by simulating various downsampling rates on the offline data. Finally, the scheduler policy is optimized via offline reinforcement learning using these synthesized data, learning to maximize execution speed while strictly bounding state deviation. We empirically validate our method via simulation (Bigym (Chernyadev et al., 2025) and Libero (Liu et al., 2023)) and real-world experiments, achieving an overall $1.55\times$ and $2.17\times$ execution speedup while maintaining task performance of different embodied policies. Our contributions are summarized as follows:

- We formalize plug-and-play policy acceleration as a CMDP to maximize execution efficiency while preserving policy performance.

- We solve this CMDP by introducing world model-based state deviation as a surrogate safety constraint, enabling offline optimization of the external scheduler via counterfactual trajectory evaluation.

- We empirically validate our method across simulation benchmarks (Libero, Bigym) and real-world robotic platforms, achieving an overall $1.55\times$ and $2.17\times$ speedup while maintaining task performance of different embodied policies.

## 2. Background

### 2.1. Action Chunking in Embodied Policies

Modern embodied architectures, such as ACT (Zhao et al., 2023a), DP (Chi et al., 2025) and various VLAs (Zitkovich et al., 2023; Kim et al., 2024; Black et al., 2024), have adopted action chunking as a standard paradigm to mitigate compounding errors and improve inference efficiency. Formally, an action chunk is defined as a finite-length sequence of consecutive actions. Let $n$ denote the chunk length. At each time step $t$, given the current observation $o_t$, a embodied policy $\pi_{base}$ does not predict a single action but rather a chunk of $n$ future actions:

$$A_t = (a_t, a_{t+1}, \ldots, a_{t+n-1}) \quad (1)$$

While effective for fine-grained manipulation, this approach suffers from the high temporal redundancy of human demonstrations. The resulting "step-by-step" execution often leads to sluggish robot behavior, preventing the system from reaching its maximum hardware performance.

### 2.2. Action Downsampling for Acceleration

To optimize execution efficiency, action downsampling is employed to reduce the number of physical steps required to complete the task by decimated or interpolating the predicted action chunk. Formally, given a chunk $A_t$ and a downsample rate $k$, action downsampling will produce an accelerated sequence $A_t^k$ of reduced length $l = \lfloor n/k \rfloor$. The specific operation depends on the policy's control mode (Lynch & Park, 2017):

$$A_t^k = \begin{cases} (a_{t+k-1}, \ldots, a_{t+lk-1}) & \text{Abs} \\ (m(a_{t:t+k}), \ldots, m(a_{t+lk-k:t+lk})) & \text{Delta} \end{cases} \quad (2)$$

where $m$ is an action-merging function (e.g., summation). This is semantically reasonable in position-based action spaces because actions directly represent target waypoints; skipping intermediate steps thus maintains the original intent of the trajectory (Shi et al., 2023). However, in real-world robotic control, the actual state reached by low-level controllers rarely coincides perfectly with the target waypoint. Downsampling exacerbates this error by increasing the distance between commanded targets, making it even more difficult to ensure consistency with the original expert trajectory. These deviations in the visited states can accumulate, ultimately leading to task failure (Xu et al., 2020).

## 3. The Foundation of SuP

This section formulates plug-and-play speedup as CMDP where a scheduler policy optimizes execution efficiency subject to world model-estimated state deviation constraints.

### 3.1. Plug-and-Play Speedup via Scheduler Policy

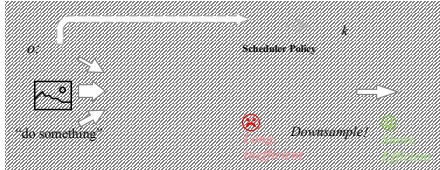

*Figure 1.* **Plug-and-Play Speedup via Scheduler Policy.** The scheduler policy $\pi$ predicts a downsampling rate $k$ to downsample the action chunk from the frozen policy into a shorter chunk for acceleration.

**Base Policy:** The base policy $\pi_{\text{base}}$ is the policy to be accelerated, which is an visumotor policy capable of predicting action chunk from visual observations and robot states. Specifically, during the inference phase of $\pi_{\text{base}}$, the input consists of the visual observation $I_t$ and robot state $o_t$, and the base policy outputs an action chunk via: $A_t \sim \pi_{\text{base}}(\cdot|I_t, o_t)$. Since $\pi_{\text{base}}$ is typically trained on slow demonstration data, the actions predicted by $\pi_{\text{base}}$ are generally inefficient in execution.

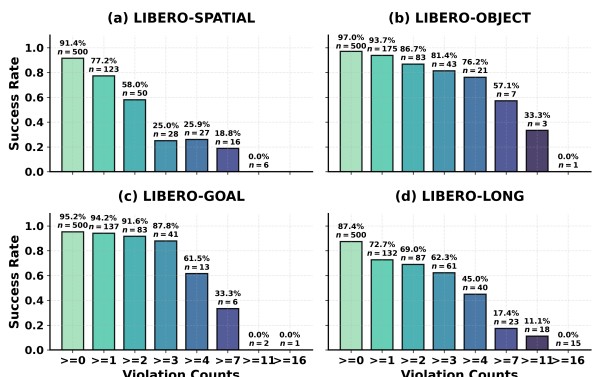

*Figure 2.* **Success rate and Violation count.** Each subplot (a–d) illustrates the relationship between the cumulative count of violations ($h_\mathcal{E} = 1$) and the task success rate across different LIBERO suites. The bars represent the conditional success rate for the subset of trajectories containing at least $x$ violations, with the sample size of each subset annotated above the corresponding bar.

**Scheduler Policy:** We introduce an additional plug-and-play scheduler policy to downsample the action chunk produced by $\pi_{\text{base}}$ for acceleration, as shown in Fig.1. Concretely, the scheduler policy $\pi(\cdot|o_t, A_t)$ is a lightweight policy that predicts a downsample rate $k$ given current state $o_t$ and action chunk $A_t$. The final action executed in the environment is thus the downsampled action chunk $A_t^k$ (Eq. 2). Specifically, when $k = 1$, the downsampled action chunk coincides with the original, i.e., $A_t^1 = A_t$. This formulation allows our scheduler $\pi$ to achieve state-dependent execution speedup of $\pi_{\text{base}}$ in a plug-and-play manner.

### 3.2. Acceleration via Constrained MDP

We then formulate speedup learning as a scheduler policy optimization problem within the framework of Constrained Markov Decision Processes (CMDPs), defined by the tuple $(\mathcal{S}, \mathcal{K}, \mathcal{P}, r, c, h, \gamma)$. In this formulation, policy acts as a high-level scheduler that optimizes execution efficiency without compromising task performance. $\mathcal{S}$ is the state space augmented to include the current environment observation $o_t$ and the action chunk $A_t$ produced by the base policy $\pi_{\text{base}}$ (i.e., $s_t = (o_t, A_t)$). $\mathcal{K}$ is the action space of our scheduler, defined as a discrete set of downsampling rates $\{k_{\min}, \ldots, k_{\max}\}$. $\mathcal{P}$ represents the environment dynamics under the execution of the downsampled action chunk $A_t^k$. To incentivize efficiency, we define the reward function as the acceleration gain: $r(s_t, k_t) = k_t$.

Ideally, the cost function $c$ for acceleration should directly reflect the impact of acceleration on task performance. We define the performance-based cost as:

$$c_q(s_t, k_t) = Q^{\pi_{\text{base}}}(o_t, A_t^k) - Q^{\pi_{\text{base}}}(o_t, A_t), \quad (3)$$

where $Q^{\pi_{\text{base}}}(o, A)$ represents the expected success rate starting from state $o$ and executing action chunk $A$

from $\pi_{\text{base}}$. The violation function is then $h_q(s_t, k_t) = \mathbb{I}[c_q(s_t, k_t) < 0]$. We provide theoretical guarantee as follows:

**Proposition 3.1.** *Given zero-violation constraint* $(h_q(s_t, k_t) = 0)$ *at each state, the scheduler is guaranteed to maintain or improve the success rate of the base policy. See Appendix A for the proof.*

Therefore, the objective of scheduler is to maximize acceleration gain under a zero-violation constraint:

$$\begin{aligned} \max_{\pi} \quad & \mathbb{E}_\pi \left[ \sum_{t=0}^{T} \gamma^t r(s_t, k_t) \right] \\ \text{s.t.} \quad & \mathbb{E}_\pi \left[ \sum_{t=0}^{T} \gamma^t h_q(s_t, k_t) \right] = 0 \end{aligned} \quad (4)$$

However, evaluating $c_q$ requires the true value function $Q^{\pi_{\text{base}}}$, which is expensive to estimate online. To enable offline learning, we transition from value-based constraints to state-based constraints using a learned world model.

### 3.3. World Model based State Deviation as Cost

To evaluate costs offline, we utilize a learned world model $\mathcal{M}_\theta$ (Sec. 4.1) to simulate future trajectories. Our key insight is that the success of $\pi_{\text{base}}$ is tied to its specific motion intent; thus, if the accelerated trajectory remains close to the original one, the task performance is preserved.

Formally, let $\tau_t = \{o_{t+1}, \ldots, o_{t+n}\}$ and $\tau_t^k$ be the sequence of states predicted by $\mathcal{M}_\theta$ under the original chunk $A_t$ and the downsampled chunk $A_t^k$. We denote $\hat{\tau}_t^k = \{\hat{o}_{t+1}^k, \ldots, \hat{o}_{t+n}^k\}$ as the version of $\tau_t^k$ temporally interpolated to match the length of $\tau_t$. We then define the state deviation $\mathcal{E}$ as the maximum discrepancy between the end-effector (EEF) of these two trajectories:

$$\mathcal{E}(s_t, k_t) = \max_{i \in [1,n]} d(\text{EEF}(o_{t+i}), \text{EEF}(\hat{o}_{t+i}^k)), \quad (5)$$

where $d(\cdot, \cdot)$ is a distance metric for EEF (Appendix F.2).

We empirically validate the reliability of our metric by evaluating the $\pi_{0.5}$ model (Black et al., 2025a) on the LIBERO benchmark with a fixed downsampling rate of $k = 2$. As shown in Fig. 2, our analysis across four task suites reveals a consistent trend: as the number of violations-defined as $h_\mathcal{E}(s_t, k_t) = \mathbb{I}[\mathcal{E}(s_t, k_t) > \epsilon]$—within a trajectory increases, the policy's success rate exhibits an evident decline. This pronounced negative correlation serves as strong evidence that state deviation is a faithful proxy for execution risk, validating its effectiveness as a cost signal.

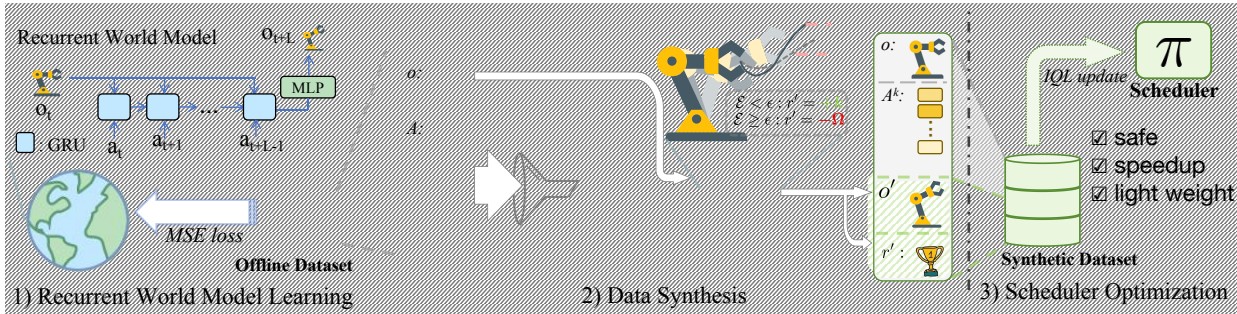

*Figure 3.* **The training process of SuP.** Our method is trained purely on offline datasets through three phases: (1) recurrent world model learning; (2) data synthesis; and (3) scheduler optimization via IQL. Through this pipeline, we optimize the scheduler to achieve the maximum possible speedup while preserving comparable performance.

# 4. Practical Implementation

To solve the CMDP for adaptive manipulation acceleration, we propose Speedup Patch (SuP), a dual-stage framework. Our objective is to learn an optimal scheduler policy $\pi_\phi$ that maximizes execution speed while maintaining task fidelity from an offline demonstration dataset $\mathcal{D}$. We first train a recurrent world model (RWM) on $\mathcal{D}$ (Sec. 4.1), allowing us to evaluate potential state deviations and constraint violations. Using the RWM as a data evaluator and generator, we synthesize a CMDP dataset. We then optimize the final scheduler via Implicit Q-Learning (Sec. 4.2).

## 4.1. Recurrent World Model

To estimate the state deviation $\mathcal{E}$ and the violation signal $h$ in an offline setting, we develop a recurrent world model (RWM), denoted as $\mathcal{M}_\theta$. To ensure the framework remains lightweight, $\mathcal{M}_\theta$ is designed to predict the robot's proprioceptive state $o$ directly, thereby bypassing the high-dimensional reconstruction of visual observations $I$. We intentionally do not use image observations as direct inputs to the RWM. Since our deviation metric is defined on executed robot trajectories, especially end-effector motion, low-dimensional proprioceptive states are sufficient for risk estimation in our setting. We further verify in Sec. 5.4 that adding visual features does not improve, and can slightly degrade, both world-model quality and downstream SuP performance.

Our RWM architecture is built upon ADM (Lin et al., 2024; 2026), which is specifically capable of handling variable-length action sequences while mitigating the compounding errors typical in multi-step rollouts. A pivotal feature of this design is that predicted states $\hat{o}$ are never fed back as inputs for subsequent time steps. Instead, the model evolves its hidden state exclusively through the action sequence.

The RWM is optimized via variable-length sequence supervision. During training, we sample action sequences $A_t$ of variable length $L \in [1, L_{max}]$ from the offline dataset $\mathcal{D}$, where the range of $L$ is chosen to encompass the temporal

scales generated by different downsampling factors $k$. The parameters $\theta$ are updated to minimize the multi-step mean squared error:

$$\mathcal{L}(\theta) = \mathbb{E}_{(o_t, A_t, o_{t+1:t+L}) \sim \mathcal{D}} \left[ \sum_{i=1}^{L} \|\hat{o}_{t+i} - o_{t+i}\|_2^2 \right]. \quad (6)$$

## 4.2. Scheduler Optimization

With the world model $\mathcal{M}_\theta$, we now describe the learning process for the scheduler policy $\pi_\phi$. To solve the CMDP, we first transform the constrained problem into an unconstrained MDP via a penalty-augmented reward:

$$r'(s, k) = \begin{cases} k, & h_\mathcal{E}(s, k) = 0 \\ -\Omega, & h_\mathcal{E}(s, k) = 1 \end{cases} \quad (7)$$

where $\Omega$ is a sufficiently large penalty to guarantee zero-violation safety (Prop. 4.1). To facilitate offline training, we construct a synthetic dataset $\mathcal{D}'$ by re-labeling the original demonstrations. Specifically, for each transition in $\mathcal{D}$, we use $\mathcal{M}_\theta$ to evaluate the violation signal $h$ across all potential downsampling factors $k$, generating a rich set of counterfactual transitions $(o, A^k, r', o')$.

**Proposition 4.1.** *Let $K_{\max}$ be the maximum possible speedup rate and $\gamma \in [0, 1)$ be the discount factor. If the penalty $\Omega$ satisfies the condition:*

$$\Omega > \frac{\gamma K_{\max}}{1 - \gamma}, \quad (8)$$

*then the optimal policy $\pi^*$ maximizing the cumulative reward of $r'(s, k)$ satisfies the constraint $h(s, \pi^*(s)) = 0$ for all reachable states $s$. See Appendix B for the proof.*

In a standard RL setting, computing temporal difference (TD) errors requires the next action $A'$, which is unavailable since we do not know how expert will behave with $s'$. We resolve this by transforming the policy input: instead of directly processing the raw tuple $(s, k)$, the scheduler maps the action chunk $A$ and skip-length $k$ into a downsampled

**Algorithm 1** Training Procedure of SuP
- 1: **Input:** Offline demonstration dataset $\mathcal{D}$, minimum and maximum downsampling rate $k_{\min}, k_{\max}$ penalty $\Omega$, deviation threshold $\epsilon$.
- 2: **Output:** Scheduler policy $\pi_\phi$.
- 3: {**Phase 1: Recurrent World Model Learning**}
- 4: Initialize world model $\mathcal{M}_\theta$.
- 5: **while** not converged **do**
- 6:     Sample batch of data $(o_t, A_t, o_{t+1:t+L})$ from $\mathcal{D}$.
- 7:     Update $\mathcal{M}_\theta$ with Eq. 6.
- 8: **end while**
- 9: {**Phase 2: Data Synthesis**}
- 10: Initialize synthetic dataset $\mathcal{D}' \leftarrow \emptyset$.
- 11: **for** each transition $(o_t, A_t)$ in $\mathcal{D}$ **do**
- 12:     **for** $k = k_{\min}$ to $k_{\max}$ **do**
- 13:         Construct downsampled action chunk $A_t^k$.
- 14:         Predict next states $\hat{o}'_{t+1:t+L} \leftarrow \mathcal{M}_\theta(o, A^k)$.
- 15:         Estimate deviation $\mathcal{E}$ with Eq. 5 and violation signal $h_\mathcal{E} \leftarrow \mathbb{I}(\mathcal{E} > \epsilon)$.
- 16:         Compute reward $r'_t$ with Eq. 7.
- 17:         Store transition $(o_t, A_t^k, r'_t, \hat{o}'_{t+L})$ into $\mathcal{D}'$.
- 18:     **end for**
- 19: **end for**
- 20: {**Phase 3: Scheduler Optimization via IQL**}
- 21: Initialize IQL networks $V_\psi, Q_\phi$.
- 22: **while** not converged **do**
- 23:     Sample batch $(o, A^k, r', o')$ from $\mathcal{D}'$.
- 24:     Update $V_\psi, Q_\phi$ with Eq. 9.
- 25: **end while**
- 26: **Return** Scheduler $\pi_\phi(o, A) = \arg\max_k Q_\phi(o, A^k)$.

representation $A^k$. This yields a semi-MDP transition from $(o, A^k)$ to $o'$ that remains Markovian with respect to the current observation and executed macro-action, while preserving the executed action sequence and reward semantics. By conditioning the policy on $(o, A^k)$, we can treat the resulting $s'$ directly as the subsequent state in a Markovian transition, effectively bypassing the need for future action chunks during value estimation.

Finally, we employ implicit q-learning (IQL) (Kostrikov et al., 2021) to optimize the scheduler on $\mathcal{D}'$. The value function $V_\psi$ and Q-function $Q_\phi$ are learned via expectile regression:

$$
\begin{aligned}
L_Q(\phi) &= \mathbb{E}_{(o, A^k, r', o') \sim D'}[(r' + \gamma V_\psi(o') - Q_\phi(o, A^k))^2], \\
L_V(\psi) &= \mathbb{E}_{(o, A^k) \sim D'}[L_2^\alpha(V_\psi(o) - Q_\phi(o, A^k))],
\end{aligned}
\tag{9}
$$

where $L_2^\alpha(x) = |\alpha - \mathbb{I}(x < 0)|x^2$ is the expectile loss. During inference, the optimal skip-length is determined by $\pi_\phi(o, A) = \arg\max_k Q_\phi(o, A^k)$.

# 5. Experiments

In this section, we conduct extensive experiments to evaluate the effectiveness of the proposed SuP framework. Specifically, we aim to investigate: (1) whether SuP can achieve significant execution speedup while preserving task success rates (Sec. 5.1); (2) the versatility of SuP across different embodied architectures of $\pi_{\text{base}}$ (Sec. 5.1); (3) SuP's empirical performance and reliability in real-world robotic experiments (Sec. 5.2); (4) the mechanism by which SuP dynamically selects appropriate downsampling ratios across diverse task scenarios (Sec. 5.3); and (5) the individual contributions of RWM, IQL, and different deviation threshold $\epsilon$ settings to the overall system performance (Sec. 5.4). See Appendix G for more analysis of SuP.

## 5.1. Simulation Task Experiments

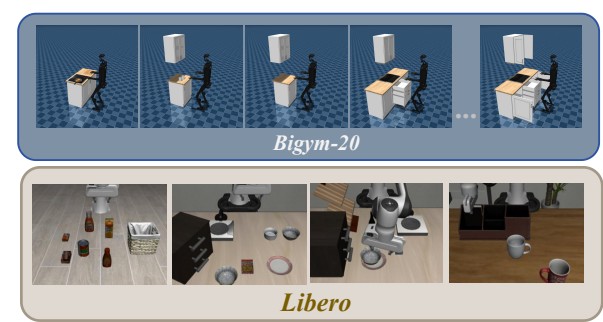

*Figure 4.* **Simulation Tasks.** We systematically evaluate SuP across 20 tasks from Bigym and 4 task suites (40 tasks in total) from Libero.

**Compared methods.** For the baseline methods considered in our comparative experiments, we select the following approaches: **Vanilla Downsample(-ds\*)**, which applies a fixed downsampling rate to all action chunks and **Demo-Speedup**, which speedup expert demonstration data with entropy estimation to retrain $\pi_{\text{base}}$. In contrast, **SAIL** achieves adaptive speedup through policy and system-level optimizations.

**Task Setup.** For simulation tasks, We validate the SuP algorithm with two benchmarks: **Bigym** (Chernyadev et al., 2025), a humanoid robot with kitchen/household manipulation that requires precise control and scene comprehension; **Libero** (Liu et al., 2023), a robotic arm grasping benchmark for VLA models, covering 4 task suite, where policies need strong instruction-following abilities. For $\pi_{\text{base}}$, we evaluate our framework across diverse architectures: ACT (Zhao et al., 2023a) and DP (Chi et al., 2025) trained on task-specific expert demonstrations for Bigym; pre-trained $\pi_{0.5}$ (Black et al., 2025a) and VLA-Adapter (Wang et al., 2025) (weights can be directly downloads from Internet) for Libero. Regarding the scheduler training, we train task-specific SuP schedulers for each Bigym environment, while for Libero, a single scheduler is trained for each task suite.

*Table 1.* SuP speedup results on Bigym Tasks compared with baselines. Each cell in the table reports two metrics: the success rate followed by the average steps to completion, where only successful trajectories are counted. Higher success rates and lower step counts indicate better performance. Cells highlighted in orange denote the best success rate for each task, while those in gray indicate a performance **drop exceeding 5%** compared to the best success rate. Best result in each column is bolded. 5 task results are shown, see **all task results in Appendix** C.

| | Method | Sandwich Remove | Take Cups | Put Cups | Drawers Close All | (15 more tasks) . . . | Cupboards Close All | Average |
|---|---|---|---|---|---|---|---|---|
| ACT | -base | 0.45, 340.5 | 0.15, 288.3 | 0.28, 320.3 | 1.0, 100.0 | . . . | 1.0, 449.8 | 0.66, 1.00× |
| | -ds2 | 0.48, 186.3 | 0.13, 178.0 | 0.36, 175.9 | 1.0, 52.0 | . . . | 1.0, 234.0 | 0.61, 1.65× |
| | +*DemoSpeedup* | 0.56, 171.5 | 0.10, 183.9 | 0.33, 169.3 | 1.0, 54.0 | . . . | 1.0, **202.1** | 0.61, 2.21× |
| | +*SuP(Ours)* | **0.64**, 155.9 | **0.20**, 176.6 | **0.38**, 156.0 | 1.0, **40.0** | . . . | 1.0, 212.1 | **0.67**, **2.01**× |
| DP | -base | 0.40, 376.8 | 0.07, 284.6 | **0.28**, 307.7 | 0.66, 118.9 | . . . | 0.90, 544.0 | 0.51, 1.00× |
| | -ds2 | 0.40, 209.6 | 0.12, 218.0 | 0.22, 181.5 | 0.54, 64.9 | . . . | 0.75, 270.3 | 0.40, 1.42× |
| | +*SAIL* | 0.42, 182.1 | 0.06, 199.3 | 0.22, 188.5 | 0.32, 82.8 | . . . | 0.88, 201.1 | 0.41, 2.01× |
| | +*DemoSpeedup* | 0.35, 199.3 | **0.21**, 239.5 | 0.25, 143.9 | 0.37, 49.2 | . . . | 0.60, 231.0 | 0.46, 1.99× |
| | +*SuP(Ours)* | **0.42**, 179.9 | 0.19, 203.0 | 0.27, 200.8 | 0.66, **65.2** | . . . | **0.91**, 146.2 | **0.51**, 1.48× |

*Table 2.* SuP speedup results on Libero compared with baselines. Each cell in the table reports two metrics: the success rate followed by the average steps to completion, where only successful trajectories are counted. Higher success rates and lower step counts indicate better performance. Cells highlighted in orange denote the best success rate for each task, while those in gray indicate a performance **drop exceeding 1%** compared to the best success rate.

| | Method | Spatial | Long | Goal | Object | Average |
|---|---|---|---|---|---|---|
| $\pi_{0.5}$ | -base | **0.988**, 105.3 | 0.924, 267.9 | 0.980, 113.1 | 0.982, 138.1 | 0.969, 1.00× |
| | -ds2 | 0.914, 67.9 | 0.874, 153.4 | 0.952, 67.6 | 0.970, 75.0 | 0.928, 1.72× |
| | +*DemoSpeedup* | 0.964, 88.1 | 0.932, 221.4 | 0.968, 88.7 | 0.988, 114.1 | 0.963, 1.22× |
| | +*SuP(Ours)* | 0.972, 70.4 | **0.940**, 215.2 | **0.986**, 93.4 | **0.994**, 83.0 | **0.973**, **1.35**× |
| VLA-Adapter | -base | **0.922**, 99.6 | **0.936**, 255.1 | **0.970**, 107.0 | 0.942, 136.2 | **0.942**, 1.00× |
| | -ds2 | 0.802, 57.1 | 0.834, 147.6 | 0.930, 57.1 | 0.882, 76.3 | 0.862, 1.77× |
| | +*DemoSpeedup* | - | - | - | - | - |
| | +*SuP(Ours)* | 0.912, 77.3 | 0.934, 204.6 | 0.956, 74.2 | **0.944**, 91.4 | 0.937, **1.34**× |

**Metrics.** To evaluate performance, we report the success rate and the average episode length of successful rollouts as a measure of efficiency. We conduct 100 evaluation trials for each task in BiGym, and 500 trials per task suite in Libero. More details of simulation can be found in App. D.

**Speedup Performance.** The main experimental results on Bigym are presented in Tab. 1, and those on Libero are summarized in Tab. 2. Across both challenging benchmarks, SuP demonstrates a superior capability to accelerate inference while maintaining, and often enhancing performance. Unlike baselines such as standard downsampling (-ds2), SAIL and DemoSpeedup, which frequently suffer from performance degradation (evidenced by the gray cells indicating a noticeable performance drop), SuP consistently maintains the original performance $\pi_{\text{base}}$. On Bigym, SuP achieves substantial average speedups (e.g., 2.01× for ACT) while maintaining its success rate. Similarly, on Libero, SuP yields a 1.35× speedup for $\pi_{0.5}$ with a peak average success rate of 0.973, effectively decoupling inference speed from performance loss and proving its robustness in numerous simulation tasks. Such gains are not purely accidental: moderate acceleration can increase effective episode progress

under fixed episode budgets and can occasionally induce more favorable interaction pacing.

**Universality across Architectures.** SuP exhibits strong generalizability across diverse policy backbones, ranging from ACT and DP to VLAs. For the ACT architecture, SuP not only doubles the inference speed but also improves the average success rate compared to the base policy. Crucially, on DP, which is sensitive to temporal modifications, SuP successfully mitigates the severe performance collapse observed in other acceleration methods (where -ds2 drops success to 0.4), recovering the success rate to 0.51 with a 1.48× speedup. This consistent efficacy extends to VLA architectures while Demospeedup faces **compatibility issues**, which is inapplicable to VLA-Adapter due to the architecture's lack of support for entropy estimation. In contrast, SuP outperforms naive downsampling strategies on both VLA-Adapter and $\pi_{0.5}$, delivering stable acceleration without compromising decision-making precision.

**Computational Efficiency.** As shown in Tab. 4, SuP achieves high computational efficiency in both training and inference time. Unlike DemoSpeedup, which incurs high

computational costs by training on the massive parameters of VLA models and requiring frequent base policy queries, SuP is exceptionally lightweight with only 5.12M trainable parameters. Crucially, our training strategy completely decouples policy learning from base policy inference; instead of querying $\pi_{base}$, we optimize our light-weight world model and scheduler using only offline data. This design drastically reduces training time, and the scheduler's inference overhead (1 ms) is negligible compared with the 50 ms inference latency of the $\pi_{0.5}$.

## 5.2. Real-world Task Experiments

To evaluate the practical efficacy of SuP in physical environments, we deployed SuP on a dual-arm robotic platform, similar to the aloha setup (Zhao et al., 2023a), focusing on manipulation tasks with multiple steps that require a balance between execution speed and operational success. Our evaluation suite consists of three tasks (Fig. 5): *Arrange Table*, *Fold Towel* and *Stack Plates*. Among these, *Arrange Table* and *Stack Plates* are conducted as single-arm setting, while *Fold Towel* serves as a bimanual task involving deformable object manipulation. Detailed descriptions of the experimental hardware and task setup are provided in Appendix E.

As summarized in Tab. 3, the results demonstrate that SuP consistently maintains high success rates while achieving significant temporal speedups across all tasks. Specifically, SuP achieves an average speedup of $2.17\times$ over the base policy $\pi_{0.5}$, outperforming the $2.07\times$ of Demospeedup while simultaneously maintaining the original success rate. This performance stands in sharp contrast to naive acceleration strategies; as highlighted in gray, the aggressive ds3 strategy leads to a catastrophic collapse in manipulation capability, dropping the success rate to 0.356. Notably, in the challenging bimanual *Fold Towel* task, which requires precise coordination for deformable objects, SuP attains the highest success rate and the lowest step count, validating its robustness in improving real-world task efficiency.

## 5.3. Case Study

To analyze how SuP dynamically selects the downsampling rate, we visualize the selected rates during the *Fold Towel* task (Fig. 6). In the plot, the blue-shaded regions correspond to phases where the model strictly predicts a low downsampling rate ($k = 2$), while red-shaded regions highlight periods of accelerated execution ($k = 4$). We observe that this behavior is highly interpretable: the model maintains the low rate during precision-critical phases such as "Approach & Contact". Conversely, during gross motion phases like Push & Move" or "Flip", the model increases the rate to exploit temporal redundancy. This demonstrates that SuP effectively distinguishes between key decision points and

translational phases, accelerating execution without sacrificing control where it matters most.

## 5.4. Ablation Study

**Impact of ADM-based world model.** To validate the effectiveness of our design, we compare our ADM-based world model against a standard MLP baseline on the Libero benchmark. As illustrated in Fig. 7, MLP model exhibits inconsistent, non-monotonic behavior in the Spatial suite, failing to correctly associate high violation counts with task failure. In contrast, ADM maintains a monotonic decrease in success rates as violations increase. Furthermore, quantitative analysis across all suites confirms that ADM achieves consistently stronger negative Spearman correlations (Spearman, 1961), demonstrating its superior capability in capturing the inverse relationship between safety violations and task success.

**Effect of Visual Inputs in the World Model.** A natural question is whether the world model should additionally take image observations as input, especially in tasks requiring spatial reasoning. To this end, we compare proprioception-only and vision-augmented variants (DINOv2 (Oquab et al., 2023) encoder) on LIBERO-Spatial. As shown in Tab. 5, adding visual features does not improve either world-model quality or downstream SuP performance. In both architectures, the proprioception-only variant achieves lower MAE, stronger violation-success correlation, and slightly better success rate and acceleration ratio. These results suggest that, in our setting, simple visual augmentation is not beneficial, so we adopt the more efficient proprioception-only design. However, it does not imply that scene information is unimportant. Rather, they indicate that robust multimodal fusion is still nontrivial for offline risk estimation in our setting. We provide an additional comparison with DreamerV3 (Hafner et al., 2023) in Appendix G.1, which shows a similar trend.

**Impact of IQL.** We compare SuP with the MPC baseline, which greedily select the highest downsample rate without violation $h_\epsilon$. As shown in Tab. 6, the MPC exhibits inferior performance in both success rate and task efficiency. This is because MPC's greedy selection only considers immediate constraints, leading to cumulative errors that compromise long-term stability.

**Sensitivity on $\epsilon$.** We evaluate the performance under different deviation thresholds. As shown in Tab. 6, a small threshold $\epsilon = 0.01$ is overly conservative and suppresses beneficial acceleration, leading to sub-optimal efficiency. A large threshold $\epsilon = 0.02$ permits excessive deviations and degrades success rates. The intermediate value $\epsilon = 0.015$ achieves the best trade-off: it still allows enough acceleration to improve effective episode progress and error recovery, while keeping risky deviations under control. Appendix F.4 further discusses how $\epsilon$ can be selected offline from the

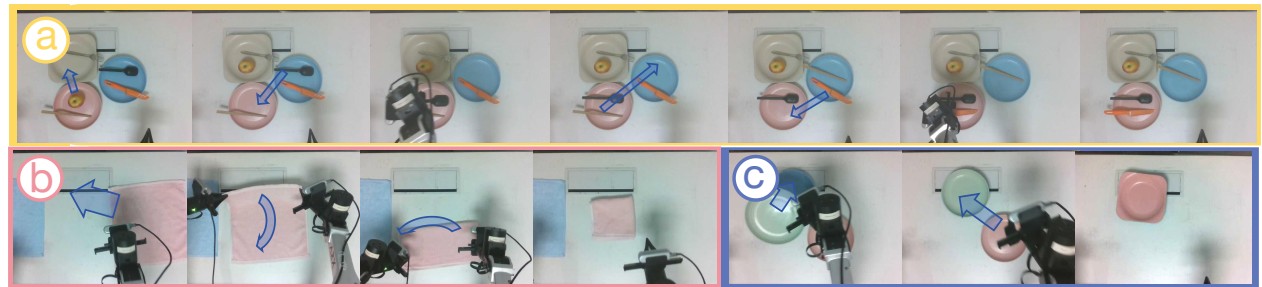

*Figure 5.* Real-world Tasks Illustration. We illustrate the procedure of 3 real-world tasks: (a) *Arrange Table* (b) *Fold Towel* (c) *Stack Plates*.

*Table 3.* SuP speedup results on real-world tasks compared with baselines. Highlighted cells follow the same convention as in Tab. 1.

| | Method | Arrange Table | Fold Towel | Stack Plates | Average |
|---|---|---|---|---|---|
| $\pi_{0.5}$ | -base | 11/30, 537.8 | 15/30, 519.5 | 27/30, 221.5 | 0.589, 1.00× |
| | +ds2 | 10/30, 291.9 | 14/30, 326.3 | 27/30, 177.1 | 0.567, 1.61× |
| | +ds3 | 3/30, 247.2 | 8/30, 187.7 | 21/30, 148.4 | 0.356, 2.19× |
| | +DemoSpeedup | 12/30, 267.4 | 14/30, 223.2 | **28/30**, 124.4 | 0.600, 2.07× |
| | +SuP(Ours) | **13/30**, 258.5 | **16/30**, 192.5 | 26/30, 138.5 | **0.611**, **2.17**× |

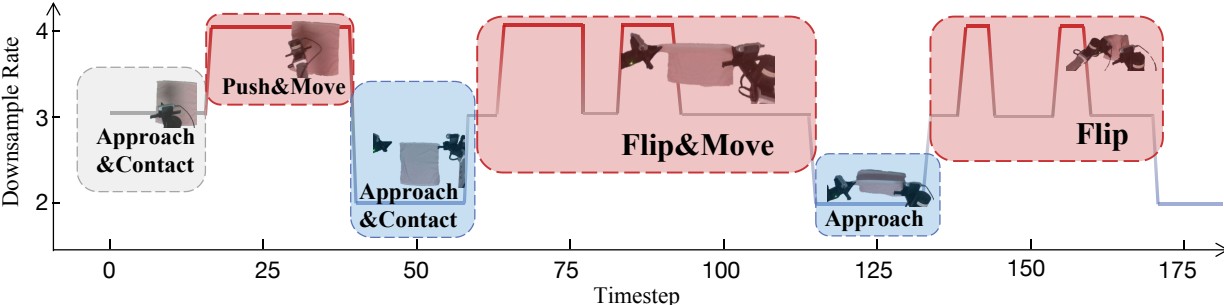

*Figure 6.* Case study. We visualize the adaptive downsampling strategy during a *Fold Towel* task. The plot tracks the predicted downsampling rate over the episode timesteps. Shaded regions annotate distinct task phases.

*Table 4.* Computational Efficiency of SuP in Libero.

| Method | Training Params | Training Time | Inference Overhead |
|---|---|---|---|
| DemoSpeedup | 4B | 20h | - |
| SuP (Ours) | 5.12M | 2h | 1ms (2%) |

*Table 5.* Effect of visual inputs on LIBERO-Spatial.

| Model | MAE | Correlation | Performance |
|---|---|---|---|
| ADM + DINOv2 | 0.013 | -0.468 | 0.964, 1.39 |
| ADM + proprio-only | 0.008 | -0.504 | 0.972, 1.50 |

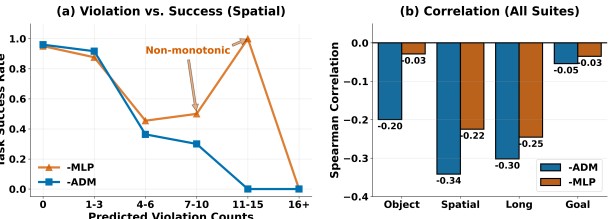

*Figure 7.* Comparison between ADM-based and MLP-based world model. (a) Visualization of task success rates against predicted violation counts in the Spatial suite. (b) Spearman correlation scores across four LIBERO suites.

dataset.

## 6. Related Works

**Imitation Learning in Embodied AI.** Imitation learning has established itself as a dominant paradigm in embodied AI (Zare et al., 2024; Ravichandar et al., 2020). The field has evolved from standard regression to sophisticated generative backbones (Chi et al., 2025; Lipman et al., 2022;

Zhang et al., 2024b) and Vision-Language-Action (VLA) models (Zitkovich et al., 2023; Kim et al., 2024; Black et al., 2024; 2025a) that adeptly handle multi-modal data and semantic understanding. Recently, world-model approaches have further advanced visual robotic manipulation (Pang et al., 2025; Zhang et al., 2024a; 2026b). However, despite their impressive success rates and generalization capabilities, these modern methods universally share a critical limitation: by faithfully mimicking the temporal pacing of human training data, they fail to fully exploit the execution

*Table 6.* Ablation study on SuP. We compare the performance impact of the IQL module and different deviation thresholds $\epsilon$.

| Ablation | Sandwich Remove | Put Cup | Long |
|---|---|---|---|
| $\pi_{base}$ | 0.45, 340.5 | 0.28, 320.3 | 0.924, 267.9 |
| SuP-0.01 | 0.59, 187.5 | 0.35, 159.8 | **0.940**, 215.2 |
| SuP-0.015 | **0.64**, 155.9 | **0.38**, 156.0 | 0.922, 191.8 |
| SuP-0.02 | 0.46, 157.6 | 0.21, 143.3 | 0.904, 168.4 |
| MPC-0.01 | 0.54, 181.4 | 0.32, 154.8 | 0.936, 225.2 |
| MPC-0.015 | 0.59, 173.3 | 0.3, 146.1 | 0.924, 205.6 |
| MPC-0.02 | 0.35, 162.1 | 0.19, 148.4 | 0.922, 173.7 |

speed potential inherent to robotic hardware.

**Fast Policy Execution.** Achieving fast policy execution is a longstanding objective in robotics, essential for deployment in dynamic real-world environments (Pham & Pham, 2019; Kiyokawa et al., 2022). Recent work handles the inference latency induced by large VLA models via asynchronous inference (Black et al., 2025b;c; Tang et al., 2025), model compression (Yang et al., 2025; Gao et al., 2025; Wu et al., 2025) or other tricks (Ma et al., 2025). While these methods accelerate individual decision cycles, they overlook the *temporal redundancy* inherent in control horizons, which has also been studied through semantic temporal abstraction for efficient RL (Liu et al., 2025). Consequently, recent research has explored reducing the total execution steps (trajectory shortening). These approaches generally fall into data-centric strategies that retrain policies on downsampled demonstrations (Guo et al., 2025; Kim et al., 2025), or model-centric methods introducing adaptive skipping modules (Arachchige et al., 2025; Yuan et al., 2025). However, such methods typically mandate computationally expensive retraining or risky online exploration. In contrast, SuP enables plug-and-play acceleration purely from offline data, eliminating these overheads.

## 7. Conclusion

We present Speedup Patch (SuP), a framework that enables plug-and-play execution acceleration for embodied policies using solely offline data. SuP employs a World Model to simulate potential outcomes of accelerated actions, allowing the CMDP-based scheduler to strategically skip redundant time steps without compromising task success rate. Our experiments on simulation and real-world tasks demonstrate that SuP consistently achieves significant speedups across diverse architectures without sacrificing success rates.

SuP also has several limitations. Currently, SuP uses a proprioception-centered world model for risk estimation. While this design is efficient and reliable in our benchmarks, it may miss hazards that are only visible through rich scene observations, especially in cluttered or highly dynamic en-

vironments. Moreover, SuP is currently restricted to discrete downsample rates. Future work can move beyond action chunk downsampling-based speedup and design a more flexible, non-integer speedup mechanism to achieve finer-grained speedup, or explore extension to multi-agent embodied systems (Yuan et al., 2023; Feng et al., 2026).

## Acknowledgement

This work was supported by the National Natural Science Foundation of China under Grants 62495090, 62495093, 62506159, U24A20324, and the Natural Science Foundation of Jiangsu under Grants BK20241199, and the "111 Center" (No. B26023), Fundamental and Interdisciplinary Disciplines Breakthrough Plan of the Ministry of Education of China (No. JYB2025XDXM118)

## Impact Statement

This paper presents work whose goal is to advance the field of Machine Learning. There are many potential societal consequences of our work, none of which we feel must be specifically highlighted here.

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

# A. Proof for Proposition 3.1

We aim to prove that: $\forall s, Q^{\pi}(s, A^k) \geq Q^{\pi}(s, A)$ (zero-violation), implies $V^{\pi'}(s_0) \geq V^{\pi}(s_0)$ (success rate guarantee). In the following derivation, we assume $\gamma = 1$.

First, we must rigorously define what the "Action Chunk" $A$ and its Q-value represent in terms of atomic, low-level control actions. Let an action chunk $A$ consist of a sequence of atomic actions $u$ over a physical duration $L$.

- Original Chunk $A$: Sequence $\{u_1, u_2, \ldots, u_L\}$.

- Accelerated Chunk $A^k$: Sequence $\{u'_1, u'_2, \ldots, u'_{L'}\}$ where $L' = L/k$.

The execution of a chunk is not a single jump, but a trajectory of atomic state transitions. Let $s_{\tau,i}$ denote the state at the $i$-th atomic step within the execution of chunk $A_{\tau}$ (where $\tau$ is the chunk index). The Q-value of a chunk $A$ under policy $\pi$ is defined as the sum of atomic rewards within the chunk plus the value of the state after the chunk finishes:

$$Q^{\pi}(s_{\tau}, A) = \mathbb{E}\left[\underbrace{\sum_{i=1}^{L} r(s_{\tau,i}, u_i)}_{\text{Intra-chunk Reward}} + V^{\pi}(s_{\tau+1})\right], \tag{10}$$

where $s_{\tau+1}$ is the state reached after executing the last atomic action $u_L$. Similarly, for the accelerated chunk $A^k$:

$$Q^{\pi}(s_{\tau}, A^k) = \mathbb{E}\left[\sum_{j=1}^{L'} r(s'_{\tau,j}, u'_j) + V^{\pi}(s'_{\tau+1})\right]. \tag{11}$$

We now prove that $V^{\pi'}(s) \geq V^{\pi}(s)$ for all states, which directly implies the final required inequality. We use mathematical induction (or recursive expansion) over the sequence of chunks. Let $V^{\pi'}(s)$ be the value of following the scheduler policy $\pi'$ (which always selects $A^k$). By definition:

$$V^{\pi'}(s_0) = Q^{\pi'}(s_0, A_0^k) = \mathbb{E}\left[\sum_{j=1}^{L'_0} r(s'_{0,j}, u'_j) + V^{\pi'}(s_1)\right]. \tag{12}$$

We now show that $V^{\pi}(s_0) \leq V^{\pi'}(s_0)$ with recursive expansion of the value function:

$$V^{\pi}(s_0) = Q^{\pi}(s_0, A_0) \leq Q^{\pi}(s_0, A_0^k) \tag{13}$$

$$= \mathbb{E}_{s_1 \sim \pi'}\left[\sum_{j=1}^{L'_0} r(s'_{0,j}, u'_{0,j}) + V^{\pi}(s_1)\right] \tag{14}$$

$$= \mathbb{E}_{s_1 \sim \pi'}\left[\sum_{j=1}^{L'_0} r(s'_{0,j}, u'_{0,j}) + Q^{\pi}(s_1, A_1)\right] \tag{15}$$

$$\leq \mathbb{E}_{s_1 \sim \pi'}\left[\sum_{j=1}^{L'_0} r(s'_{0,j}, u'_{0,j}) + Q^{\pi}(s_1, A_1^k)\right] \tag{16}$$

$$= \mathbb{E}_{s_1, s_2 \sim \pi'}\left[\sum_{j=1}^{L'_0} r(s'_{0,j}, u'_{0,j}) + \sum_{m=1}^{L'_1} r(s'_{1,m}, u'_{1,m}) + V^{\pi}(s_2)\right] \tag{17}$$

$$\vdots \tag{18}$$

$$\leq \mathbb{E}_{\tau \sim \pi'}\left[\sum_{t=0}^{\infty}\sum_{j=1}^{L'_t} r(s'_{t,j}, u'_{t,j})\right] = V^{\pi'}(s_0). \tag{19}$$

**Remarks on** $\gamma = 1$: We emphasize that the undiscounted setting ($\gamma = 1$) is both physically motivated and mathematically essential for our derivation. First, since our primary metric is the task success rate under sparse rewards, setting $\gamma = 1$ ensures that the value function $V^\pi(s)$ directly represents the probability of success, i.e., $V^\pi(s) = \mathbb{P}(\text{Success}|\pi, s)$. Second, $\gamma = 1$ is a necessary condition for the validity of the telescoping sum in Eq. 19. In a variable-duration setting where the scheduler accelerates execution, the physical arrival time at any future state $s_{t+1}$ differs from that of the base policy. If $\gamma < 1$, the discount factors associated with $s_{t+1}$ would not align between the two policies, preventing the intermediate terms in the performance difference expansion from canceling out. By assuming $\gamma = 1$, the value of a state becomes invariant to $L$ and $L'$, allowing for a rigorous proof of global performance preservation despite temporal downsampling.

## B. Proof for Proposition 4.1

Let $Q^*(s, k)$ denote the optimal action-value function. The maximum possible value of a safe trajectory is bounded by $V_{\max} = \sum_{t=0}^{\infty} \gamma^t K_{\max} = \frac{K_{\max}}{1-\gamma}$. Consider an arbitrary state $s$. If an action $k_v$ violates the constraint (i.e., $h(s,k) = 1$), its Q-value is bounded by:

$$Q^*(s, k_v) = -\Omega + \gamma \mathbb{E}[V^*(s')] \leq -\Omega + \gamma V_{\max}. \tag{20}$$

Conversely, since valid actions yield at least a reward of 1 (assuming $k \geq 1$), the Q-value of the optimal safe action $k_{safe}$ satisfies $Q^*(s, k_{safe}) \geq 1 + \gamma$. To ensure the optimal policy never selects a violation, we require $Q^*(s, k_{safe}) > Q^*(s, k_v)$. It suffices to show:

$$1 + \gamma > -\Omega + \frac{\gamma K_{\max}}{1-\gamma} \implies \Omega > \frac{\gamma K_{\max}}{1-\gamma} - 1 - \gamma. \tag{21}$$

Thus, setting $\Omega > \frac{\gamma K_{\max}}{1-\gamma}$ (a strictly stronger condition) guarantees that any violating action has a lower value than any valid action, compelling the optimal policy to strictly satisfy the safety constraint.

## C. Whole Results of Bigym

We report the (Success Rate, Episode Length) pairs for all 20 tasks across both ACT and DP architectures. As shown in Tab. 7, SuP achieves the best balance between efficiency and success rate in most tasks, outperforming both static downsampling and DemoSpeedup baselines across a wide range of manipulation skills.

## D. Simulation Experiment Detail

### D.1. Bigym

Here, we provide details of the BiGym tasks: we utilize a total of 20 tasks, all set in a kitchen scenario. The task descriptions (which can serve as language prompts if required) are listed below: (1) Sandwich Remove: Take the sandwich out of the frying pan. (2) Take Cups: Take two cups out from the closed wall cabinet and put them on the table. (3) Put Cups: Pick up cups from the table and put them into the closed wall cabinet. (4) Dishwasher Open Trays: Pull out the dishwasher's trays with the door initially open. (5) Move Plate: Move the plate between two draining racks. (6) Saucepan to Hob: Take the saucepan from the closed cabinet and place it on the hob. (7) Flip Cutlery: Take the cutlery from the static holder, flip it, and place it back into the holder. (8) Cupboards Close All: Close all drawers and doors of the kitchen set. (9) Sandwich Flip: Flip the sandwich in the frying pan using the spatula. (10) Dishwasher Close Trays: Push the dishwasher's trays back with the door initially open. (11) Pick Box: Pick up a large box from the floor and place it on the counter. (12) Drawers Close All: Close all sliding drawers of the kitchen cabinet. (13) Drawers Open All: Open all sliding drawers of the kitchen cabinet. (14) Dishwasher Close: Push back all trays and close the door of the dishwasher. (15) Wall Cupboard Open: Open doors of the wall cabinet. (16) Store Box: Move a large box from the counter to the shelf in the cabinet below. (17) Wall Cupboard Close: Close doors of the wall cabinet. (18) Dishwasher Open: Open the dishwasher door and pull out all trays. (19) Sandwich Toast: Use the spatula to put the sandwich on the frying pan and toast it. (20) Flip Cup: Flip the cup initially positioned upside down on the table to an upright position.

### 1. Observation Space (State Space)
BiGym's observation space is hybrid, combining visual inputs, proprioceptive data, and (for bi-manual mode) base state, which is defined as:

$$O = \{I_{\text{head}}, I_{\text{left}}, I_{\text{right}}, s_{\text{proprio}}\}$$

*Table 7.* Performance comparison of different methods in all Bigym tasks. Best success rates and shortest lengths are **bolded**. Method with best success rate per task is highlighted.

| | Method | Sandwich Remove | Take Cups | Put Cups | Dishwasher Open Trays | Move Plate |
|---|---|---|---|---|---|---|
| ACT | -base | (0.45, 340.5) | (0.15, 288.3) | (0.28, 320.3) | (1.0, 275.0) | (**0.58**, 194.8) |
| | -ds2 | (0.48, 186.3) | (0.13, 178.0) | (0.36, 175.9) | (1.0, 169.0) | (0.50, 155.0) |
| | +*DemoSpeedup* | (0.56, 171.5) | (0.10, 183.9) | (0.33, 169.3) | (1.0, 156.0) | (0.28, **71.7**) |
| | +*SuP(Ours)* | (**0.64**, **155.9**) | (**0.20**, 176.6) | (**0.38**, 156.0) | (1.0, 131.0) | (0.49, 167.2) |
| DP | -base | (0.40, 376.8) | (0.07, 284.6) | (**0.28**, 307.7) | (0.57, 351.2) | (0.33, 261.1) |
| | -ds2 | (0.40, 209.6) | (0.12, 218.0) | (0.22, 181.5) | (0.48, 310.2) | (0.36, 203.3) |
| | +*SAIL* | (0.42, 182.1) | (0.06, **199.3**) | (0.22, 188.5) | (0.96, 177.2) | (0.38, **108.7**) |
| | +*DemoSpeedup* | (0.35, 199.3) | (**0.21**, 239.5) | (0.25, **143.9**) | (0.94, **108.8**) | (0.38, 178.8) |
| | +*SuP(Ours)* | (**0.42**, 179.9) | (0.19, **203.0**) | (0.27, 200.8) | (**0.97**, 267.4) | (**0.39**, 164.8) |

| | Method | Saucepan to Hob | Flip Cutlery | Cupboards Close All | Sandwich Flip | Dishwasher Close Trays |
|---|---|---|---|---|---|---|
| ACT | -base | (0.78, 334.1) | (0.35, 243.6) | (1.0, 449.8) | (0.20, 406.7) | (1.0, 200.3) |
| | -ds2 | (0.47, 248.7) | (0.22, 257.6) | (1.0, 234.0) | (0.19, 239.4) | (1.0, 117.0) |
| | +*DemoSpeedup* | (0.78, **154.5**) | (**0.39**, 90.8) | (**1.0**, 202.1) | (0.18, 156.8) | (1.0, 95.0) |
| | +*SuP(Ours)* | (**0.88**, 174.3) | (0.27, 138.7) | (1.0, 212.1) | (**0.22**, 194.5) | (**1.0**, 86.0) |
| DP | -base | (**0.69**, 426.1) | (0.06, 368.0) | (0.90, 544.0) | (0.06, 436.0) | (0.94, 210.4) |
| | -ds2 | (0.58, 285.0) | (0.12, 356.7) | (0.75, 270.3) | (0.03, 157.3) | (0.53, 146.4) |
| | +*SAIL* | (0.58, 217.8) | (0.10, **120.7**) | (0.88, 201.1) | (0.06, 311.9) | (0.26, 165.5) |
| | +*DemoSpeedup* | (0.60, **157.8**) | (0.10, **145.9**) | (0.60, 231.0) | (0.11, 294.5) | (**0.96**, **108.3**) |
| | +*SuP(Ours)* | (0.66, 332.7) | (**0.16**, 389.9) | (**0.91**, 146.2) | (**0.11**, 311.5) | (0.72, 175.4) |

| | Method | Pick Box | Drawers Close All | Drawers Open All | Dishwasher Close | Wall Cupboard Open |
|---|---|---|---|---|---|---|
| ACT | -base | (0.25, 372.0) | (1.0, 100.0) | (1.0, 325.5) | (1.0, 175.0) | (1.0, 148.8) |
| | -ds2 | (0.04, 182.0) | (1.0, 52.0) | (0.99, 190.0) | (1.0, 90.9) | (1.0, 74.5) |
| | +*DemoSpeedup* | (0.02, 173.0) | (1.0, 54.0) | (0.99, 163.0) | (**1.0**, 83.9) | (0.92, 64.0) |
| | +*SuP(Ours)* | (0.25, **317.9**) | (**1.0**, 40.0) | (**1.0**, 152.2) | (1.0, 84.0) | (**1.0**, 65.0) |
| DP | -base | (0.0, -) | (0.66, 118.9) | (**0.89**, 478.1) | (**0.99**, 178.9) | (0.89, 260.0) |
| | -ds2 | (0.0, -) | (0.54, 64.9) | (0.12, 412.7) | (0.54, **64.9**) | (0.88, **131.0**) |
| | +*SAIL* | (0.0, -) | (0.32, 82.8) | (0.48, 207.9) | (0.76, 91.1) | (0.84, **111.2**) |
| | +*DemoSpeedup* | (0.0, -) | (0.37, **49.2**) | (0.44, **193.2**) | (0.9, 158.4) | (**0.91**, 147.2) |
| | +*SuP(Ours)* | (0.0, -) | (**0.66**, 65.2) | (0.30, 442.9) | (0.95, 147.3) | (**0.91**, 146.2) |

| | Method | Store Box | Wall Cupboard Close | Dishwasher Open | Sandwich Toast | Flip Cup |
|---|---|---|---|---|---|---|
| ACT | -base | (0.51, 454.4) | (1.0, 100.0) | (1.0, 389.0) | (0.08, 596.9) | (**0.53**, 312.7) |
| | -ds2 | (0.41, 229.2) | (1.0, 52.0) | (1.0, 377.0) | (0.07, 261.9) | (0.32, 182.4) |
| | +*DemoSpeedup* | (0.33, **222.2**) | (**1.0**, 46.0) | (1.0, 163.3) | (0.09, **157.1**) | (0.30, **149.8**) |
| | +*SuP(Ours)* | (**0.53**, 231.2) | (1.0, 54.0) | (1.0, 157.4) | (**0.13**, 171.2) | (0.46, 181.0) |
| DP | -base | (0.25, 456.0) | (1.0, 96.0) | (0.57, 354.1) | (0.04, 426.0) | (0.01, 312.0) |
| | -ds2 | (0.39, 328.6) | (1.0, 57.5) | (0.56, 281.3) | (0.03, **178.7**) | (0.03, 552.0) |
| | +*SAIL* | (0.06, 296.0) | (1.0, 51.9) | (0.38, 244.3) | (0.02, 209.0) | (0.04, 155.4) |
| | +*DemoSpeedup* | (0.14, **296.0**) | (**1.0**, **47.5**) | (0.38, **113.5**) | (0.01, 180.0) | (0.04, **150.3**) |
| | +*SuP(Ours)* | (**0.39**, 336.6) | (1.0, 50.6) | (**0.57**, 267.4) | (**0.07**, 240.9) | (**0.05**, 164.5) |

- **Visual Observations**: RGB images ($I_{head}, I_{left}, I_{right}$) from three cameras (forehead, left wrist, right wrist), with a default resolution of $84 \times 84$.

- **Proprioceptive State ($s_{proprio}$)**: The state space adopts the Bi-manual mode, with the low-dimensional state ranging from 60 to 70 dimensions, including joint angles, joint velocities, and base states (where the leg control is configured in floating base mode), etc.

## 2. Action Space
The action space $A \in \mathbb{R}^{16}$ in Bigym can be formularized as three parts:

$$A = \{A_{arms}(\mathbb{R}^{10}), A_{base}(\mathbb{R}^4), A_{grip}(\mathbb{R}^2)\},$$

where $\{A_{arms}$ controls the qpos of the robot arm, $A_{base}$ controls the floating base (i.e. legs) of the robot and $A_{grip}$ controls the left and right gripper of the robot arm.

## 3. Training of $\pi_{base}$

*Table 8.* Hyperparameters for ACT and DP in Bigym.

| ACT Hyperparameters | | DP Hyperparameters | |
|---|---|---|---|
| Hyperparameter | ACT | Hyperparameter | DP |
| Learning Rate | 1e-5 | Learning Rate | 1e-4 |
| Weight Decay | 1e-4 | Weight Decay | 1e-6 |
| Batch Size | 64 | Batch Size | 64 |
| Chunk Size ($k$) | 24 | Observation Horizon | 2 |
| Feedforward Dim | 3200 | Action Horizon | 24 |
| Hidden Dim | 512 | Diffusion Steps | 100 |
| Encoder Layers | 4 | Noise Scheduler | DDPM |
| Decoder Layers | 7 | Kernel Size | 5 |
| Attention Heads | 8 | Vision Model | MVT (Seo et al., 2023) |
| Dropout | 0.1 | Down Dims | [256,512,1024] |

In the Bigym environment, we followed the implementation of ACT and DP from the DemoSpeedup's open-source repository (specifically the robobase folder)[1]. Due to the absence of pre-released checkpoints, we retrained the algorithms across 20 environments according to the original source code. We selected the models that achieved the highest win rates during evaluation as our base policies. The training hyperparameters for ACT and DP are detailed in Tab. 8, respectively. Our experiments revealed that the performance of DP in Bigym was generally inferior to that of ACT. We also attempted to train a model based on $\pi_{0.5}$, but we found that the success rates were lower than those of both ACT and DP in a lot of tasks. This suggests that the model may not be suitable for whole-body control tasks. Consequently, we did not attempt to accelerate the VLA base policy in Bigym.

### D.2. Libero

The Libero suite comprises four specialized sub-suites, each designed to isolate or integrate specific types of knowledge transfer for robot manipulation tasks, with distinct focuses and standardized language instruction patterns. Libero-spatial focuses on the transfer of declarative knowledge about spatial relationships, using instructions that specify spatial descriptors and target objects; Libero-object targets declarative knowledge about object concepts, with instructions centered on object names and containers; Libero-goal concentrates on procedural knowledge about task goals, featuring instructions that outline action-oriented tasks; Libero-long consists of long-horizon tasks involving entangled declarative and procedural knowledge transfer, with multi-step instructions that combine spatial, object, and goal concepts. Below is a detailed breakdown of the observation and action spaces common to or specific to each sub-suite, along with their core characteristics.

**1. Observation Space (State Space)**
Libero's observation space is hybrid, combining visual inputs, proprioceptive data, which is defined as:

$$O = \{I_{\text{top}}, I_{\text{wrist}}, s_{\text{proprio}}\}$$

- **Visual Observations**: RGB images ($I_{\text{top}}, I_{\text{wrist}}$) from two cameras (top, wrist), with a default resolution of $224 \times 224$.

- **Proprioceptive State ($s_{\text{proprio}}$)**: The state space is an 8-dimensional low-dimensional joint state space.

**2. Action Space**
The action space is 7-dimensional, with 6-dimensional delta-EEF control, and 1-dimensional Gripper control.

**3. Detail of $\pi_{\text{base}}$**
In the Libero environment, we utilized the officially released pre-trained model checkpoints. For $\pi_{0.5}$, we obtained the corresponding model parameters by adhering to the instructions provided in its open-source repository[2]. Similarly, for the VLA-Adapter, we followed the instructions outlined in its respective repository[3]. Specifically, $\pi_{0.5}$ employs a shared set

---

[1] https://github.com/lingxiao-guo/DemoSpeedup/tree/main/robobase
[2] https://github.com/Physical-Intelligence/openpi/tree/main/examples/libero
[3] https://github.com/OpenHelix-Team/VLA-Adapter

of model weights across all four task suites, whereas the VLA-Adapter utilizes independent weights for each suite. The training demonstration data is downloaded directly via HuggingFace[4].

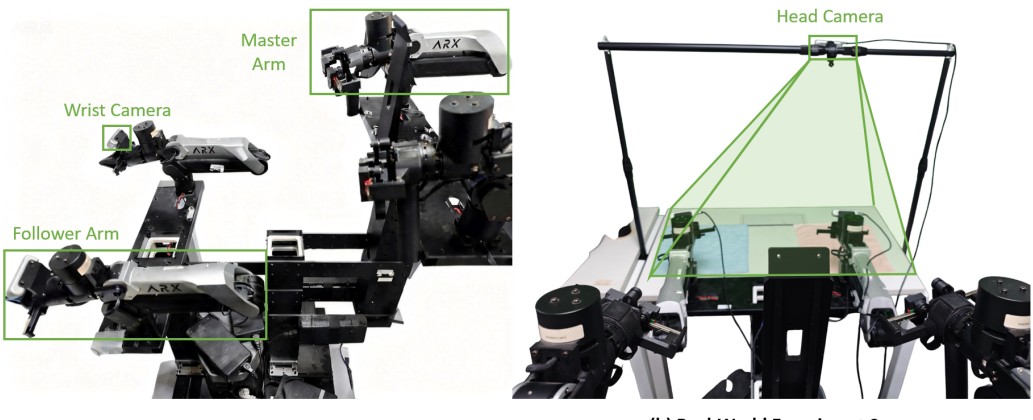

**(a) ARX Master-Follower Design**   **(b) Real-World Experiment Scene**

*Figure 8.* ARX5 illustration. (a) The master-follow design for data collection (b) The actual scene of our real-world experiment.

## E. Real-world Experiment Detail

### E.1. Hardware Setup

The hardware configuration is detailed in Fig. 8. We utilize the ARX5 robotic platform, a dual-arm system analogous to Aloha, consisting of two master arms and two puppet arms. Both arms were actively employed for dual-arm teleoperation and data collection. To provide visual feedback, a top-mounted RealSense D435i camera captures the RGB image observations required for the experiments.

### E.2. Details of Real-world Tasks

**Fold Towel.** The scene consists of two towels of different colors or patterns placed on the tabletop. The robot is required to identify the target towel specified by a linguistic instruction and execute a folding sequence. This task tests the policy's ability to handle deformable objects and its grounding of language instructions in a multi-object scene.

**Arrange Table.** This task involves three plates and five objects initially distributed across them (arranged in a 2, 2, 1 pattern). The robot must follow a three or four-step instruction to pick and place specific objects into designated plates. This task represents a long-horizon challenge requiring precise spatial reasoning and high-level planning.

**Stack Plates.** Three plates are placed separately on the table. The robot must stack them into a single pile following a specific order provided in the instruction (e.g., bottom-to-top sequence). This task emphasizes contact-rich manipulation and the strict maintenance of operational order.

**Training of $\pi_{\text{base}}$.** We collected a total of 200 high-quality demonstrations using teleoperation, with 50 trajectories in Fold Towel, 100 trajectories in Arrange Table and 50 trajectories in Stack Plates. We then utilized the $\pi_{0.5}$ model as the foundation. The model was fine-tuned on task-specific trajectoris to serve as $\pi_{\text{base}}$, ensuring reliable execution of the fundamental manipulation primitives.

## F. More Details for SuP

In this section, we provide the detailed implementation of SuP, including how to downsample gripper action, how to calculate state deviation and the architecture of Recurrent World Model and scheduler.

---

[4]https://huggingface.co/datasets/openvla/modified_libero_rlds

## F.1. Gripper Action Compensation for Downsampling

Our methods rely on action chunk downsampling strategy that remain semantically aligned with the original one. While the downsampling strategy described in Sec. 2.2 ensures that the robot's arm waypoints remain spatially consistent in the sense of they desired, gripper actions require separate consideration due to their binary nature and specific physical constraints. In most simulation environments, gripper actions are represented as binary signals (e.g., $< 0$ for closed, $> 0$ for open). Regardless of whether absolute or relative position control is used, standard downsampling causes a mismatch in the cumulative physical displacement of the gripper. For example, if a full grasp requires several consecutive closure commands, reducing the action frequency results in the gripper failing to reach the intended state in time, leading to failed grasps.To resolve this inconsistency and maintain the success rate after downsampling, we applied the following task-specific compensations:

- BiGym: We followed the method described in DemoSpeedup (Guo et al., 2025) by increasing the control gain of the gripper, ensuring it responds more aggressively to the reduced number of commands.

- Libero: We doubled the magnitude (velocity) of each gripper action command.

For instance, if a single original action resulted in a 0.1 cm closure, the adjusted action for a downsampling factor of 2 ($N = 2$) produces a 0.2 cm closure. Although this adjustment is specifically tailored for a downsampling rate of 2, we found it to be a highly effective heuristic for maintaining physical state consistency. The necessity of gripper action compensation is quantitatively validated in Tab. 9. Without the fix, naive downsampling ($N = 2$) leads to a significant performance degradation, with the average success rate dropping from 96.9% to 84.2%, particularly in the libero-spatial task where the gripper often fails to secure objects due to insufficient closure displacement. By applying our proposed compensation—adjusting the gripper's response magnitude—the "-ds2" variant recovers the average success rate to 92.6% while maintaining a high inference speedup ($1.72\times$). This results in a much more robust balance between efficiency and task reliability.

*Table 9.* Ablation study of gripper action compensation on Libero benchmarks. We compare the original policy ($pi_{0.5}$) with downsampled versions ($N = 2$) before and after applying the gripper fix.

| Method | spatial | long | goal | object | Average |
|---|---|---|---|---|---|
| $\pi_{0.5}$ (Original) | 0.988, 105.3 | 0.924, 267.9 | 0.980, 113.1 | 0.982, 138.1 | 0.969, $1.00\times$ |
| No Grip Fix (-ds2) | 0.708, 77.0 | 0.818, 167.4 | 0.888, 66.2 | 0.954, 84.4 | 0.842, $1.58\times$ |
| With Grip Fix (-ds2) | 0.914, 67.9 | 0.874, 153.4 | 0.952, 67.6 | 0.970, 75.0 | 0.928, $1.72\times$ |

## F.2. Calculation of State Deviation

In this section, we detail calculation of the state deviation metric $\mathcal{E}$, which serves as the core criterion for the switching logic within our Speedup Patch (SuP) framework. The calculation of state deviation relies on the formal representation of the robot's spatial configuration via the End-Effector (EEF) pose. The EEF pose is defined as a combination of its 3D Cartesian coordinates $(x, y, z)$ and its orientation, represented internally as a unit quaternion to avoid singularities. To evaluate the fidelity of the robot's motion during downsampled execution with a rate $k$, we determine the "expected" state at any intermediate sub-step $i \in \{1, \ldots, k-1\}$ through pose interpolation between two consecutive reference waypoints $e_t$ and $e_{t+k}$ produced by the base policy. Specifically, the reference position is obtained via linear interpolation, while the reference orientation is computed using Normalized Linear Interpolation (NLERP) (Shoemake, 1985). This approach ensures that the interpolated orientation remains on the unit hypersphere by normalizing the result of a linear interpolation between the two reference quaternions, providing a computationally efficient approximation of the shortest rotation path.

To evaluate the fidelity of the generated trajectories, we define a composite distance metric $d(e_{curr}, e_{ref})$ that measures the discrepancy between the current and reference end-effector (EEF) states. This distance comprises two components: the Euclidean distance for translational position and the geodesic distance for rotational orientation. The total distance at step $t + i$ is formulated as:

$$d_{t+i} = \frac{1}{2}\sqrt{(x_{curr} - x_{ref})^2 + (y_{curr} - y_{ref})^2 + (z_{curr} - z_{ref})^2} + \cdot \arccos(|\langle q_{curr}, q_{ref}\rangle|)$$

where $\mathbf{p} = [x, y, z]^\top$ represents the Cartesian coordinates and $q$ denotes the orientation expressed as a unit quaternion. The rotational term calculates the minimum angular displacement between the two orientations, using the absolute value of the inner product $\langle q_{curr}, q_{ref} \rangle$ to account for the antipodal property of quaternions.

From an implementation perspective, these geometric operations—including NLERP and geodesic distance calculations—are natively and efficiently supported by the *scipy.spatial.transform.Rotation module* in the SciPy library.

### F.3. Network architecture

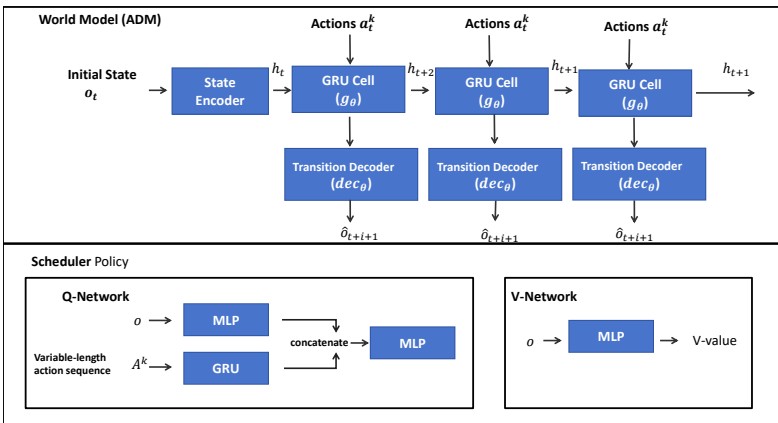

*Figure 9.* Network architecture of Recurrent World Model and Scheduler Policy.

**Recurrent World Model.** The Any-Step Dynamics Model (ADM) is designed to predict future trajectories while bypassing the recursive error accumulation typical of auto-regressive transitions. The process begins by mapping the initial observation $o_t$ to a latent representation $h_t = \text{enc}_\theta(o_t)$ using a state encoder. This latent vector serves as the initial hidden state for a Gated Recurrent Unit (GRU), denoted as $g_\theta$. For each step $i \in \{0, \dots, L-1\}$, the GRU updates the hidden state via $h_{t+i+1} = g_\theta(h_{t+i}, a_{t+i}^k)$, conditioned on the previous latent state and the external action $a_{t+i}^k$. Crucially, a transition decoder $\text{dec}_\theta$ maps each latent state directly to a predicted observation $\hat{o}_{t+i+1}$. By decoupling the latent dynamics from the observation space—specifically by ensuring predicted observations are never fed back as inputs—the model maintains high trajectory fidelity and provides a stable foundation for counterfactual evaluation.

**Scheduler Policy.** The scheduler policy is implemented within the Implicit Q-Learning (IQL) framework, comprising separate Q and V networks. To handle the variable-length nature of the action sequences $A^k$, the Q-network employs a dual-stream architecture: a GRU processes the temporal dependencies of the action sequence, while a standard Multi-Layer Perceptron (MLP) encodes the current environment state $o$. The resulting features are concatenated and passed through a secondary MLP to produce the final Q-value. In contrast, the V-network utilizes a simplified architecture, consisting of a single MLP that maps the environment state $o$ directly to a state-value estimate. This design ensures the policy can effectively evaluate complex, multi-step action plans against the current environmental context.

*Table 10.* Deviation thresholds $\epsilon$ corresponding to different target safe-to-unsafe ratios on the LIBERO task suites.

| Ratio | Spatial | Object | Goal | Long | Average |
|-------|---------|--------|------|------|---------|
| 0.33 | 0.0107 | 0.0099 | 0.0138 | 0.0106 | 0.0112 |
| 1 | 0.0150 | 0.0140 | 0.0196 | 0.0159 | 0.0162 |
| 3 | 0.0217 | 0.0193 | 0.0285 | 0.0244 | 0.0235 |

### F.4. Selection of deviation threshold $\epsilon$ offline

In this section, we describe an offline procedure for choosing the deviation threshold $\epsilon$. The value of $\epsilon$ determines the ratio of safe to unsafe transitions used to train the scheduler policy, and thus indirectly controls the achievable acceleration ratio. We therefore choose $\epsilon$ offline by matching a desired safe-to-unsafe ratio. Tab. 10 reports $\epsilon$ for different target ratios on the

LIBERO task suites.

## G. More Analysis of SuP

### G.1. More Vision Input Ablation Results

We additionally compare proprioception-only and vision-augmented variants under DreamerV3 on LIBERO-Spatial (Tab. 11). Consistent with the ADM results in the main text, adding DINOv2 features does not improve downstream performance. Specifically, DreamerV3 with proprioception-only input achieves slightly lower MAE (0.009 vs. 0.010), stronger violation-success correlation ($-0.494$ vs. $-0.491$), higher success rate (0.972 vs. 0.970), and a higher acceleration ratio (1.48 vs. 1.45) than the vision-augmented variant. This suggests that the lack of gains from visual input is not specific to ADM, but persists in other world-model architectures in our setting as well. We attribute this to the difficulty of modality fusion.

*Table 11.* Effect of visual inputs in the world model on LIBERO-Spatial.

| Model | MAE | Correlation | Success Rate | Acc. Ratio |
|---|---|---|---|---|
| ADM + DINOv2 | 0.013 | -0.468 | 0.964 | 1.39 |
| ADM + proprio-only | 0.008 | -0.504 | 0.972 | 1.50 |
| DreamerV3 + DINOv2 | 0.010 | -0.491 | 0.970 | 1.45 |
| DreamerV3 + proprio-only | 0.009 | -0.494 | 0.972 | 1.48 |

### G.2. Effect of World Model Accuracy on Downstream Performance

Since SuP relies on the recurrent world model to estimate deviation and label safe or unsafe acceleration decisions, a natural question is how world-model accuracy affects the final scheduler performance. To study this, we evaluate world models from different training stages and report their prediction error (MAE), the correlation between predicted violations and downstream task success, and the final SuP performance on LIBERO-Spatial in Tab. 12.

The results show a clear trend that better prediction accuracy leads to better downstream acceleration performance. As the world model improves from ADM-RANDOM to ADM-FINAL, the MAE decreases substantially from 0.143 to 0.008, while the violation-success correlation becomes much stronger, changing from 0.051 to $-0.504$. At the same time, the downstream performance of SuP consistently improves, with the success rate increasing from 0.924 to 0.972 and the average episode length decreasing from 98.1 to 70.4. This suggests that a more accurate world model provides more reliable estimates of risky versus safe acceleration choices, which directly benefits scheduler learning.

At the same time, the results also suggest that SuP is not overly sensitive to prediction error once the world model is sufficiently accurate. From ADM-MIDDLE to ADM-FINAL, the MAE further decreases from 0.010 to 0.008, but the correlation changes only marginally ($-0.492$ vs. $-0.504$), and the downstream improvement is correspondingly small (0.970, 73.6 vs. 0.972, 70.4). This indicates that, beyond a certain accuracy threshold, the final performance is no longer strongly affected by small reductions in prediction error. In other words, SuP does not require a perfectly accurate world model; once the model is accurate enough to reliably distinguish relatively safe from risky acceleration decisions, the downstream policy becomes much less sensitive to the remaining prediction noise.

*Table 12.* Effect of world-model quality on downstream SuP performance on LIBERO-Spatial. Performance is reported as success rate and average episode length.

| Model | MAE | Correlation | Performance |
|---|---|---|---|
| ADM-Random | 0.143 | 0.051 | 0.924, 98.1 |
| ADM-Low | 0.023 | -0.158 | 0.956, 86.2 |
| ADM-Middle | 0.010 | -0.492 | 0.970, 73.6 |
| ADM-Final | 0.008 | -0.504 | 0.972, 70.4 |

*Table 13.* Generalization results under more diverse offline training data. Performance is reported as success rate and average episode length on held-out Task 0 from each LIBERO suite.

| Method | Spatial | Object | Goal | Long | Average |
|---|---|---|---|---|---|
| Original | 1.0, 59.9 | 0.98, 101.5 | 0.98, 95.1 | 0.98, 215.2 | 0.985, 117.9 |
| SuP-Scale | 0.98, 57.4 | 0.98, 111.3 | 0.96, 90.5 | 0.96, 221.8 | 0.97, 120.3 |
| SuP-Single | 0.94, 60.5 | 0.98, 113.6 | 0.94, 93.4 | 0.90, 204.9 | 0.94, 118.1 |

### G.3. Generalization Potential of SuP

We further investigate the generalization potential of SuP under more diverse offline training data. Instead of training a separate scheduler on each LIBERO suite, we additionally train a single scheduler using the combined data from all four suites and evaluate it on held-out Task 0 from each suite. We denote this setting as SUP-SCALE, and compare it with the original suite-specific training setup, denoted as SUP-SINGLE.

The results are shown in Tab. 13. Training with more diverse data improves the average success rate from 0.94 to 0.97, while maintaining comparable efficiency. In particular, SUP-SCALE achieves clear improvements over SUP-SINGLE on the Spatial, Goal, and Long suites, suggesting that the scheduler can benefit from shared structure across tasks rather than relying only on suite-specific patterns. Although the average episode length is slightly larger in some suites, the overall results indicate that increasing data diversity improves the robustness and generalization ability of SuP.

These findings suggest that SuP has the potential to benefit from larger and more diverse offline datasets, and that its scheduler can acquire transferable knowledge beyond a single task suite. This supports the promise of SuP as a more broadly applicable acceleration framework as embodied manipulation datasets continue to grow.

## H. Hyperparameter of SuP

We report $\Omega$ as a positive penalty magnitude, consistent with Eq. (7). In implementation, rewards are normalized by $k_{max}$. Therefore, the effective penalty under the original reward scale is the reported value multiplied by $k_{max}$.

*Table 14.* Hyperparameter configurations of SuP in different experiment settings.

| Hyperparameter | BiGym | Libero | Real-world |
|---|---|---|---|
| Learning rate | $3 \times 10^{-4}$ | $1 \times 10^{-4}$ | $1 \times 10^{-4}$ |
| Batch size | 512 | 512 | 512 |
| GRU hidden dimension | 256 | 256 | 256 |
| GRU layers | 3 | 3 | 3 |
| Chunk length | 24 | 10 | 20 |
| $k_{min}$ | 2 | 1 | 2 |
| $k_{max}$ | 4 | 2 | 4 |
| Epsilon ($\epsilon$) | 0.01-0.02 | 0.01-0.02 | 0.02-0.04 |
| Expectile ($\tau$) | 0.95 | 0.95 | 0.95 |
| Penalty ($\Omega$) | 5 | 2 | 1 |
| Gamma ($\gamma$) | 0.9 | 0.1 | 0.9 |

# I. Additional Visualization Results

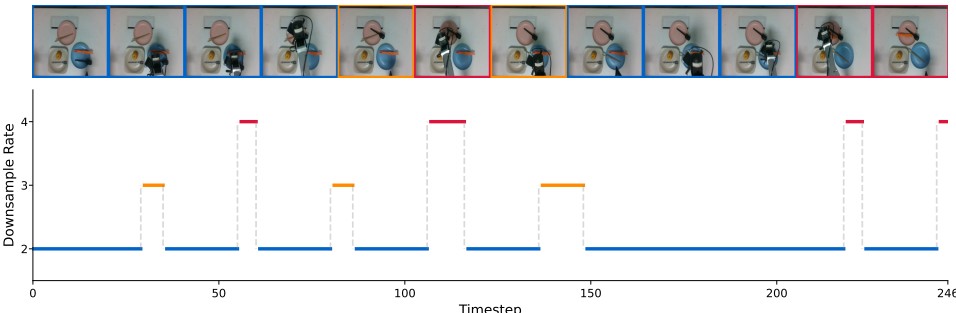

*Figure 10.* Arrange Table

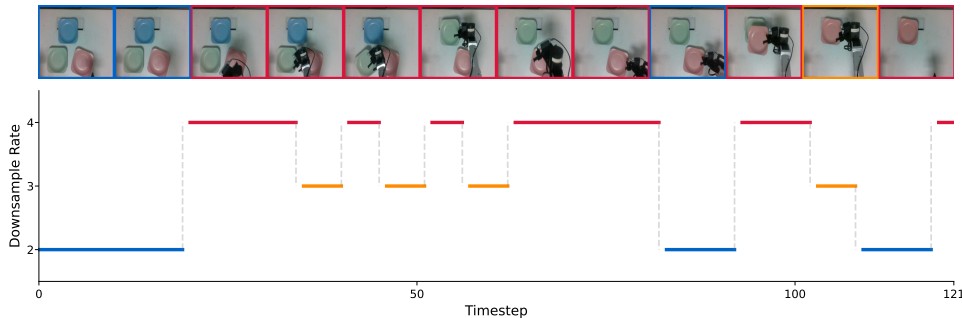

*Figure 11.* Stack Plate

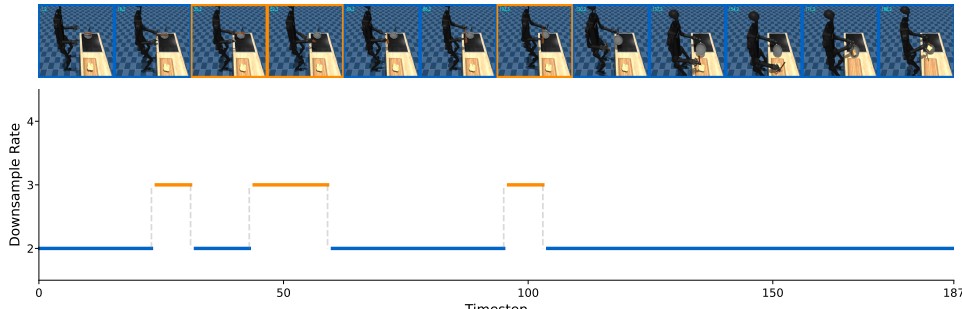

*Figure 12.* Sandwich Toast

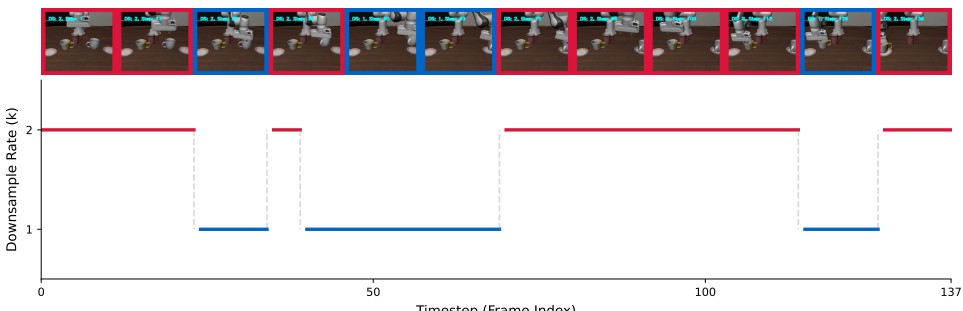

*Figure 13.* Libero Long

