# OpenReview forum: "Speedup Patch: Learning a Plug-and-Play Policy to Accelerate Embodied Manipulation"
_ICML.cc/2026/Conference — ICML 2026 regular_

### Official Review · Reviewer_2U3p · 2026-03-11

**Soundness:** 3
**Presentation:** 4
**Significance:** 4
**Originality:** 3
**Overall Recommendation:** 4
**Confidence:** 4

**Summary:**

This paper proposes Speedup Patch (SuP), a lightweight plug-and-play acceleration framework that speeds up embodied manipulation by downsampling action chunks from a frozen base policy, addressing the challenge that prior acceleration methods often require retraining or online interaction. SuP learns an external scheduler that selects a downsampling factor per action chunk and formulates this decision as a Constrained Markov Decision Process (CMDP) to maximize execution efficiency without maintaining task success. To enable fully offline learning, SuP trains a Recurrent World Model to synthesize counterfactual transitions and use a state-deviation proxy for constraint learning, optimizing the scheduler with IQL and validating consistent speedups via extensive simulation and real-robot experiments.

**Compliance With Llm Reviewing Policy:**

Affirmed.

**Final Justification:**

Final recommendation : upgrade score to 4: weak accept

The additional experimental results and clarification provided during the rebuttal phase have significantly strengthen my confidence in this work. The authors demonstrated how specific implementation components contribute to the overall performance. As the proposed method offers a valuable and easily integrable tool for the machine learning and robotics community, I have upgraded my score to reflect my positive final assessment.

**Key Questions For Authors:**

1. Penalty sign : Can the authors confirm whether the reported table values correspond to $\Omega$ or $-\Omega$ ? (Table 9, Eqs. (7) and (8))

2. MDP re-parameterization clarity : The text motivates transforming the policy input to avoid requiring future chunks during value learning. Can the paper explicitly clarify this need arises from the original state definition $s=(o, A)$ and that switching to $(o, A^k)$ makes transitions Markovian over $o$ ?

3. Interpretation of Table 5 ($\epsilon$ sensitivity)  : Does Table 5 imply that $\epsilon$ must be tuned per task/suite, and if so, what is a practical offline-only guidance for selecting $\epsilon$ without additional interaction?

4. MPC baseline implementation : Could the authors add a short appendix note specifying how MPC is implemented (e.g., planning horizon/rollout length) to ensure a fair and reproducible comparison?

**Limitations:**

yes

**Strengths And Weaknesses:**

**Strengths**

1. **Clear Motivation and Formulation** : The paper targets execution inefficiency in imitation learning-based embodied policies and frames the goal of offline-only acceleration without retraining base policies. Also, modeling speedup as a CMDP with a discrete per-chunk downsampling decision provides a simple, interpretable control variable that directly targets efficiency.

2. **Strong empirical coverage and practical validation** : The evaluation spans simulation suites and real-robot tasks, including a deformable object bimanual setting, and reports consistent speedups while retaining comparable success rates. The case study visualizes phase-dependent scheduling (low sampling rates in precision phases, while high sampling rates in gross motion), which supports interpretability of the learned scheduler behavior.

3. **Lightweight and decoupled training pipeline** : SuP is computationally efficient with a small number of trainable parameters and negligible inference overhead for the scheduler, avoiding frequent base-policy querying during training.

**Weaknesses**

1. **Formatting issue** : The manuscript exceeds the 8-page main text limit, which may violate requirements.

2. **World-model attribution is unclear (EEF-only deviation)** : The safety signal is defined via an end-effector distance metric (Eq. 5) and used as the core offline constraint. However, the world model is explicitly designed to predict the robot state $o$ while bypassing visual observation reconstruction. Therefore, it is unclear whether constraint learning captures scene/object dynamics or mostly relies on low-dimensional proprioceptive prediction. A brief clarification of what is included in $o$, and how this impacts risk estimation would strengthen the argument.

3. **CMDP vs small-penalty RL ambiguity ($\Omega$, $\gamma$)** : The paper converts the CMDP and states that $\Omega$ is sufficiently large to guarantee zero-violation (Prop. 4.1). However, the reported hyperparameters use relatively small penalty magnitudes and remarkedly different discount factors across benchmarks. This makes it unclear whether SuP is intended to enforce a hard zero-violation constraint in practice, or whether violations are allowed as a standard reward-penalty trade-off (i.e., soft constrained RL)

---

> ### Author Rebuttal · Authors · 2026-03-31
>
> We greatly appreciate the reviewer’s time and insightful suggestions. In response to the reviewer’s key concerns:
>
> >**W1,Q1,W3:** Issues about formatting and penalty $\Omega$.
>
> The $\Omega$ values in Table 9 should indeed all be positive, and the current table is misleading in this regard. The small reported magnitudes are due to reward normalization in our implementation: the acceleration reward is divided by $k_{\max}$, so the penalty coefficients are correspondingly shown on the normalized scale. Under the original scale, the effective penalty magnitudes should be the reported values multiplied by $k_{\max}$. We will correct Table 9 and explicitly clarify the normalization convention in the revised paper.
>
> >**W2:** Clarification of the world-model state $o$ and its impact on risk estimation.
>
> The $o$ denotes the robot's proprioception state, which includes the qpos and qvel of each joint, gripper closure status. In our practical implementation, since the end-effector (EEF) directly affects the gripper's positioning and final grasping performance, we set $o$ to include only EEF information instead of the full body state, to make the World Model focus more on the accuracy of EEF prediction.
>
> We agree that the absence of image observations may cause the world model to be less capable of identifying potential risks during environmental interaction. To investigate this, we performed a comparative analysis between proprioception-only and vision-augmented world models on LIBERO-Spatial, a benchmark that explicitly requires spatial reasoning capabilities. We also included Dreamer V3[1] as an additional world model architecture, which has been thoroughly investigated in pixel-based control benchmarks. Surprisingly, we observed that incorporating visual features yielded no improvement in either world model prediction accuracy or downstream task performance.
>
> |Spatial|MAE|Correlation|Success rate|Acc ratio|
> |---|---:|---:|---:|---:|
> |ADM+DINOv2|0.013|-0.468|0.964|1.39|
> |ADM+proprio-only|0.008|-0.504|0.972|1.50|
> |DreamerV3+DINOv2|0.010|-0.491|0.970|1.45|
> |DreamerV3+proprio-only|0.009|-0.494|0.972|1.48|
>
> Importantly, this degradation was far less significant with DreamerV3, which only increased MAE by 0.001. We attribute this to architectural differences in high-dimensional noise handling: DreamerV3 uses an RSSM with a stochastic latent bottleneck that filters out task-irrelevant visual noise. Despite this design, DreamerV3 still delivers no performance gains from visual input.
>
> Overall, while integrating vision is a reasonable goal, the current lack of robust modality fusion methods often causes performance degradation. Our EEF-only design thus maintains its advantage by delivering a highly efficient, noise-resistant state representation.
>
> >**Q2:** Rationale for the transformation from $(o,A),k→(o',A')$ to $o,A^k→o'$.
>
> The need for this transformation comes directly from the original state definition. Under the original formulation, the state is $(o, A)$, and choosing $k$ leads to a transition $(o,A),k→(o',A')$, so value learning must reason about the future chunk variable A'. By reformulating the decision as executing a variable-length macro-action $A^k$ from $o$, we obtain a semi-MDP with transition $o,A^k→o'$, which is Markovian with respect to the current observation and the executed macro-action. Since this reformulation preserves the executed action sequence and our reward does not depend on $A'$, it is equivalent from the optimization perspective while avoiding the need to model future chunks explicitly.
>
> >**Q3:** Interpreting Table 5 / how to choose $\epsilon$ offline
>
> Table 5 should not be interpreted as implying that $\epsilon$ must be tuned with additional interaction for every task or suite.
>
> In our experiments, the $\epsilon$ values that correspond to comparable safe:unsafe operating points are fairly consistent across LIBERO suites. This suggests that, in practice, a single default $\epsilon$ can be used for a given robot embodiment / action scale, rather than tuning separately for each task.
>
> |ratio|Spatial|Object|Goal|Long|Average|
> |---|---:|---:|---:|---:|---:|
> |0.33|0.0107|0.0099|0.0138|0.0106|0.0112|
> |1|0.0150|0.0140|0.0196|0.0159|0.0162|
> |3|0.0217|0.0193|0.0285|0.0244|0.0235|
>
> If a practitioner want to adjust the safety-versus-acceleration trade-off, $\epsilon$ can be optionally calibrated once offline using the dataset, for example by matching a target safe:unsafe ratio.
>
> >**Q4:** MPC implementation details
>
> We will clarify in the appendix that the MPC baseline uses a planning horizon of $H=1$, to match SuP’s information constraint that future action chunks are unavailable at the current step. Concretely, for each $k \in [k_{\min}, k_{\max}]$, we compute $\epsilon(s, A^k)$ with RWM and greedily choose the largest feasible $k$ satisfying $\epsilon(s, A^k) < \epsilon$.
>
> [1] Hafner, Danijar, et al. "Mastering diverse domains through world models." arXiv preprint arXiv:2301.04104 (2023).

---

> > ### Author Rebuttal · Reviewer_2U3p · 2026-04-03
> >
> > I appreciate the authors' effort in providing comprehensive responses and additional experiments. The results clearly highlight the impact of specific implementation choices and resolve the questions I had during the initial review. I would be pleased to see these results incorporated into the final version, and I have increased my score accordingly.

---

> > > ### Author Response · Authors · 2026-04-03
> > >
> > > We sincerely appreciate your feedback and are glad that the your conecrn have been fully addressed. Your questions deeply helped us improve the clarity and rigor of our work. We are very glad our responses met your expectations, and we will ensure all new insights and experiments are carefully integrated into the final version.

---

### Official Review · Reviewer_s5H4 · 2026-03-12

**Soundness:** 3
**Presentation:** 4
**Significance:** 3
**Originality:** 3
**Overall Recommendation:** 5
**Confidence:** 4

**Summary:**

This paper proposes a policy-agnostic framework (SuP) to accelerate embodied manipulation policies from offline datasets through adaptive action downsampling. The framework first trains a recurrent world model to evaluate candidate downsampling strategies and enforce safety constraints. Then, a lightweight scheduler is learned to adaptively select the downsampling rate for action chunks to improve execution speed while preserving task success rate. The proposed approach is evaluated across multiple manipulation benchmarks and real-world tasks, demonstrating consistent speedups with limited degradation in performance.

**Compliance With Llm Reviewing Policy:**

Affirmed.

**Final Justification:**

All of my concerns regarding this work, including the reliability of the learned world model, comparisons against stronger baselines, and the analysis of improved performance under intermediate rather than more conservative threshold, have been addressed in the rebuttal. Therefore, I have raised my score from 4 to 5.

**Key Questions For Authors:**

My main concern is the reliability of the world model trained from the offline manipulation dataset, as it is the key component underlying the entire proposed framework. Additional analysis or discussion on this point would strengthen the paper and increase confidence in the robustness and scalability of the approach.

**Limitations:**

Yes

**Strengths And Weaknesses:**

**Strengths:**
1. This paper targets an important problem in learning from human demonstrators: the learned policy often inherits slow teleoperation behavior and leads to suboptimal execution speed. Improving policy execution without retraining the base policy is a practical and meaningful problem.

2. The proposed framework is designed to be lightweight, plug-and-play and policy-agnostic, and does not require modifying or retraining the base policy. The scheduler can be attached to different policies and neural architectures, which is supported by experiments on Action Chunking with Transformers (ACT), Diffusion Policies (DP), and Vision-Language Action (VLA) models.

3. This work formulates the proposed objective as a constrained MDP and uses a learned world model to estimate constraint violations offline. This makes the optimization tractable without requiring expensive online interaction or retraining.

4. Experiments cover multiple benchmarks, policy architectures, and real-world manipulation tasks. The results generally support the claim that the method improves execution speed while preserving task success rates. The ablations also provide useful insights.

**Weaknesses:**
1. The proposed framework relies heavily on the world model learned from the offline manipulation dataset to estimate state deviation and enforce safety constraints. However, the paper provides limited evidence on the reliability of the learned world model predictions. Since the scheduler is trained entirely based on these predictions, errors in the world model could affect the learned downsampling policy. The world model is also trained with a relatively simple multi-step MSE objective (Lines 190–192, and Eq. 6). Recent work suggests stronger training strategies to improve prediction reliability. For example, DreamerV3 ([1], Eq. 1-3) proposes robust training recipes for latent world models, IRIS ([2], Sec. 2.2) explores sequence-modeling architectures for long-horizon dynamics prediction. Discussing these approaches or providing stronger validation of the learned world model would strengthen the paper.

2. The experiment mainly compares against fixed downsampling and DemoSpeedup. Including or discussing additional baselines (e.g. SAIL, or other methods mentioned in Introduction) will provide a clearer picture of the proposed method’s advantages.

3. In Section 5.4, item "Sensitivity on $\epsilon$", the authors claim that "a small threshold ($\epsilon$ = 0.01) results in conservative behavior and sub-optimal efficiency due to frequent RL interventions. Conversely, a large threshold ($\epsilon$ = 0.02) prioritizes speed but allows excessive deviations, leading to a significant decline in success rates...", whereas the authors' choice ($\epsilon$ = 0.015) shows not only good at efficiency, but also better performance than 0.01 in some tasks in Table 5. Could the authors provide further insights into why the intermediate threshold yields improved success rates in these cases?

4. Don't forget to update the running title.

References:

[1] Hafner, Danijar, et al. "Mastering diverse domains through world models." arXiv preprint arXiv:2301.04104 (2023).

[2] Micheli, Vincent, Eloi Alonso, and François Fleuret. "Transformers are sample-efficient world models." arXiv preprint arXiv:2209.00588 (2022).

---

> ### Author Rebuttal · Authors · 2026-03-31
>
> We are deeply grateful for the reviewer’s time and constructive suggestions. Below we provide detailed responses to your key questions and concerns:
>
> > **W1:** Concerns regarding the reliability of our MSE-trained world model. Lack of comparison with stronger architectures like DreamerV3 or IRIS.
>
> We thank the reviewer for suggesting comparisons with stronger world-model architectures such as DreamerV3 and IRIS. We conducted additional experiments with both models under the same setting on LIBERO-Spatial.
>
> |Libero-Spatial|MAE|Correlation|performance|
> |---|---:|---:|---|
> |DreamerV3|0.009|-0.494|0.972,71.2|
> |DreamerV3+TrueZ|0.006|-|-|
> |IRIS|0.015|-0.280|0.962,77.3|
> |ADM(Ours)|0.008|-0.504|0.972,70.4|
>
> These experiments lead to two observations.
>
> 1. **Stronger architectures do not provide better reliability in our setting.** DreamerV3 and IRIS do not improve over our model in MAE or downstream usefulness. We believe this is because both rely on autoregressive rollout, which can accumulate errors over the multi-step prediction horizon required by SuP.
> 2. To isolate the error source, we test **DreamerV3+TrueZ**, where rollout latents are replaced by posterior latents inferred from ground-truth observations. This reduces MAE from 0.009 to 0.006, suggesting that **rollout compounding, rather than model capacity alone, is a major source of prediction error.**
>
> Overall, these results suggest that, for SuP, reducing rollout error is more important than simply adopting a larger world-model architecture.
>
> >**W2:** Additional baselines.
>
> We have re-implemented the SAIL algorithm on the Bigym benchmark, with its core hyperparameters carefully adapted and tuned. Full results can be found in [anonymous link](https://anonymous.4open.science/r/Sup-Rebuttal-Video-1AF3).
>
> |Method|Sandwich Remove|Take Cups|Put Cups|Drawers Close All|(16 more tasks)|Cupboards Close All|Average|
> |---|---|---|---|---|---|---|---|
> |SAIL|(0.42,182.1)|(0.06,199.3)|(0.22,188.5)|(0.32,82.8)|...|(0.88,201.1)|0.41,2.01X|
> |SuP-DP(Ours)|(0.42,179.9)|(0.19,203.0)|(0.27,200.8)|(0.66,65.2)|...|(0.91,146.2)|0.51,1.48X|
>
>
> We observe that SAIL achieves an overall average success rate of 0.41 and an average speedup of 2.01×. Our analysis is as follows:
> 1. **Regarding SAIL’s high speedup rate:** it originates from the abundant free-movement phases in the kitchen arrangement task, which are categorized as non-critical and thus eligible for acceleration.
> 2. **Regarding SAIL’s degraded success rate:** SAIL’s speedup mechanism relies on predicting the action speedup factor without explicitly foreseeing and modeling how acceleration impacts future states. Consequently, aggressive acceleration at the current step might cause error accumulation, especially in scenarios involving lower-limb movements.
>
> This further highlights our SuP method: by modeling environmental transitions and integrating safe reinforcement learning, we achieve reliable acceleration while maintaining task success rate.
>
>
> >**W3:** Why the intermediate threshold ($\epsilon=0.015$) yields higher success rates than the stricter, more conservative threshold ($\epsilon=0.01$).
>
> We attribute the higher success rate of the intermediate threshold $\epsilon=0.015$ over the stricter $\epsilon=0.01$ to the beneficial effect of faster execution. A slightly looser threshold enables stronger acceleration, so the agent can make more progress earlier within the same episode budget and has more time to retry or recover from minor errors. Moreover, for some manipulation tasks, execution pacing can affect the interaction dynamics, and a slightly faster pace can be more favorable for successful completion. Therefore, the stricter $\epsilon=0.01$ threshold may over-constrain acceleration, while $\epsilon=0.015$ provides a better balance. Please see videos in this [anonymous link](https://anonymous.4open.science/r/Sup-Rebuttal-Video-1AF3) for concrete examples.
>
> >**W4:** Don't forget to update the running title.
>
> We will update the running title to “SuP: Learning a Plug-and-Play Policy to Accelerate Embodied Manipulation” in the final version.
>
> >**Q1:** Concern of the reliability of the world model trained from the offline manipulation dataset. Additional analysis or discussion is needed.
>
> We agree that world-model reliability is critical. We would like to clarify that our world model is not used as a full planner, but as a risk estimator for candidate acceleration decisions; hence, the key requirement is reliable relative ranking of safer vs. riskier choices rather than perfect long-horizon generation. Empirically, we support this reliability through (i) low prediction error, (ii) strong correlation between predicted deviation/violation and downstream performance, and (iii) additional comparisons with stronger world-model baselines (see W1), which show that our design remains competitive or better in our setting. We will revise the paper to make this motivation, evidence, and the remaining limitations clearer.

---

> > ### Author Rebuttal · Reviewer_s5H4 · 2026-04-03
> >
> > I appreciate the authors’ efforts in providing additional experiments and analysis of the learned world model, as well as comparisons with SAIL baseline. All of my concerns have been addressed, and I have raised my score to 5.

---

> > > ### Author Response · Authors · 2026-04-03
> > >
> > > We sincerely appreciate your feedback and we are glad that your concerns have been fully addressed. Your questions deeply helped us improve the clarity and rigor of our work. We will ensure that the new insights and experiments, specifically the SAIL baseline comparison and the expanded world model analysis, are carefully integrated into the appendix of the final version.

---

### Official Review · Reviewer_N6My · 2026-03-12

**Soundness:** 4
**Presentation:** 4
**Significance:** 3
**Originality:** 3
**Overall Recommendation:** 5
**Confidence:** 5

**Summary:**

This paper introduces a policy agnostic method (SuP) to speed up the execution speeds of policies, tested in simulation and in the real world. SuP consists of a scheduler optimized with offline RL to maximize downsampling rate of action chunks while being penalized for any violations achieved when executing the downsampled action chunks through a learned recurrent world model (RWM). The violation function is a distance criterion between the end-effectors states of two trajectories (predicted by the world model after executing the action chunk and a downsampled action chunk).

The RWM is a latent space world model that is trained to predict robot states, excluding the need to reconstruct expensive high dimensional visual observations. The world model is trained on the offline trajectory dataset, with trajectories resampled with different downsampling factors.

**Compliance With Llm Reviewing Policy:**

Affirmed.

**Final Justification:**

All concerns are addressed. Paper was already in pretty good shape before, just some unclear portions of the text that I trust the authors to revise appropriately.

**Key Questions For Authors:**

See weaknesses.

I overall like the paper and approach and wish to see it accepted, but critically I think some key limitations and weaknesses need to be clarified in the paper (they do not need to necessarily be fixed). With better clarifications I'm happy to raise my score.

**Limitations:**

No limitations are described. Some should be discussed e.g.
- Need for a dataset to finetune on.
- The approach is not task-agnostic and thus not truly zero-shot, and requires training a new world model and scheduler per task, requiring a new dataset per task as well. From the appendix 50-100 demos are still needed for both base model fine-tuning and the speed-up system.

**Strengths And Weaknesses:**

Strengths:
- The approach is well formulated and very well explained.
- The approach of using RL to learn downsampling rates and a world model to permit RL without having to interact with the slow-real robot is well motivated. Moreover this approach is very lightweight compared to the most related prior work DemoSpeedup, which makes it much more suitable for real world deployment.
- Figure 6 highlights the strengths of a learned, state-dependent downsampling rate scheduler, showing how the downsampling rate expectedly is much lower when there are more complex contacts, and higher when there are just free-space motions.
- The approach is policy agnostic, although it assumes there is some notion of action chunking in the policy the speed up scheduler is applied to.

Weaknesses:
- The world model is trained to predict robot states only and requires a offline dataset of trajectories (including a real dataset if speeding up real world policies). This makes the approach a little less scalable as a result.
- Another concern with the world model is that it may not be capable of capturing certain dynamics of an environment such as obstacles (e.g. the table, or a wall) as it is only predicting robot states and is not fed any data with respect to the environment. While this indeed makes the world model easier to train and fast to run, it may be a limitation when tacking more complex or cluttered environments where environment dynamics depend on more than just the robot itself.
- While the tested tasks show successful speedup of policies without major drop in performance, it is confusing how the speed up system improves success rate in most cases. Analysis on this unexpected result would be useful (e.g. where does the original policy fail that the sped up version is suddenly able to succeed in). If it's just due to noise, some confidence intervals would be helpful.
- I have some concern over the choice of tasks to test on. Libero and Bigym are both task sets with very little pose randomizations (if I remember correctly). It would be useful to highlight whether this method can handle more randomizations in e.g. object materials or more importantly object poses. One way to show this would be to show reset distributions of the real world task and simulation tasks.

---

> ### Author Rebuttal · Authors · 2026-03-31
>
> We highly appreciate your time and thoughtful remarks. We hereby clarify and respond to your concerns:
>
> >**W1:** The reliance on proprioceptive states and offline/real-world datasets limits the method's scalability.
>
> We argue that this does not restrict the scalability of our SuP method, with two key validations:
> 1. Nearly all embodied datasets include proprioceptive states synchronously collected with actions. Our method can only use offline datasets originally collected for base policy training, with no extra data cost.
> 2. We empirically verify SuP’s scalability: we additionally trained SuP with data from all four LIBERO suites and evaluated on held-out Task 0. Using more diverse data (SuP-Scale) improved generalization over suite-specific (SuP-Single) training (0.97 vs. 0.94 average success). This demonstrates that our method leverages shared cross-task knowledge to acquire more robust skills as the volume of data increases.
>
>
> |Method|Spatial|Object|Goal|Long|Average|
> |---|---|---|---|---|---|
> |Original|1.0,59.9|0.98,101.5|0.98,95.1|0.98,215.2|0.985,117.9|
> |SuP-Scale|0.98,57.4|0.98,111.3|0.96,90.5|0.96,221.8|0.97,120.3|
> |SuP-Single|0.94,60.5|0.98,113.6|0.94,93.4|0.90,204.9|0.94,118.1|
>
> >**W2:** The world model lacks environment awareness (e.g., obstacles), which may limit its capability in complex or cluttered scenes.
>
> We agree that explicit environment awareness is important in cluttered scenes, and this is a limitation of our current proprioception-only world model. To improve this, we evaluated vision-augmented world models on LIBERO-Spatial. Counter-intuitively, adding pretrained visual features (DINOv2 [1]) actually degraded world-model accuracy (increasing MAE by 0.005) and downstream SuP performance; the proprio-only model achieved comparable or better success and acceleration.
>
> |Spatial|MAE|Correlation|Success rate|Acc ratio|
> |---|---:|---:|---:|---:|
> |ADM+DINOv2|0.013|-0.468|0.964|1.39|
> |ADM+proprio-only|0.008|-0.504|0.972|1.50|
>
> Importantly, we do not interpret this as showing that scene information is unnecessary. Instead, it indicates that **simple visual feature concatenation is not sufficient for modality fusion in our current pipeline**. We will clarify this limitation in the paper, add the ablation to the appendix, and leave this as future work.
>
> >**W3:** The unexpected success rate improvement requires failure-case analysis or confidence intervals to rule out noise.
>
> We thank the reviewer for highlighting this point. We do not believe the slight improvement in success rate is merely due to statistical noise; rather, it appears to be associated with the acceleration itself. We observe two factors:
> 1. **Extended Effective Horizon & Error Recovery (e.g., LIBERO):** By accelerating execution, SuP completes tasks in less time. Within a fixed environment time limit, this grants the agent more remaining time to retry or recover from minor errors.
> 2. **Favorable Physical Dynamics via Altered Pacing (e.g., BiGym & Real-World):** The success of some tasks in manipulation is sensitive to execution pacing. We give three examples (see videos in this [anonymous link](https://anonymous.4open.science/r/Sup-Rebuttal-Video-1AF3)).
>
> >**W4:** The benchmarks lack randomization; the method's robustness to varying object poses and materials needs validation.
>
> To address the robustness concern, we additionally test robustness on two real-world tasks by increasing the randomization of the objects' initial placement. The result shows that $\pi_{base}$’s success rate drops under this harder setting, while adding SuP does not cause a clear further decrease of $\pi_{base}$. This suggests that robustness is primarily bounded by $\pi_{base}$, and SuP largely preserves rather than improves or degrades it under such shifts. See images in this [anonymous link](https://anonymous.4open.science/r/Sup-Rebuttal-Video-1AF3) for robust test illustration.
>
> ||Fold Towel|Stack Plates|
> |---|---:|---:|
> |$\pi_{base}$|9/30,527.2|19/30,235.4|
> |+SuP|8/30,194.5|20/30,142.5|
>
> >**Q1:** No limitations are described.
>
> We apologize for this omission. In the final version, we will include a dedicated "Limitations" section to discuss the current boundaries of our framework. Specifically, we will clarify: (1) its reliance on robot demonstration data; (2) the challenge in incorporating vision modality.
>
> [1] Oquab M, Darcet T, Moutakanni T, et al. Dinov2: Learning robust visual features without supervision[J]. arXiv:2304.07193, 2023.

---

> > ### Author Rebuttal · Reviewer_N6My · 2026-04-06
> >
> > All concerns largely addressed. I raise my score to a 5.

---

> > > ### Author Response · Authors · 2026-04-07
> > >
> > > We sincerely appreciate your feedback and we are glad that your concerns have been fully addressed. Your questions deeply helped us improve the clarity and rigor of our work. We will ensure that the new insights and experiments are carefully integrated into the appendix of the final version.

---

### Official Review · Reviewer_A78A · 2026-03-13

**Soundness:** 3
**Presentation:** 3
**Significance:** 3
**Originality:** 3
**Overall Recommendation:** 4
**Confidence:** 3

**Summary:**

This paper introduces a light-weight, model-agnostic method called SuP for accelerating embodied policies. The key idea of SuP is to eliminate redundancies by using an external scheduler to adaptively downsample action chunks. Empirical results on LIBERO and Bigym demonstrated a remarkable acceleration rate for diverse policies.

**Compliance With Llm Reviewing Policy:**

Affirmed.

**Final Justification:**

All of my concerns have been resolved during the rebuttal. Therefore, I raise my score from 3 to 4.

**Key Questions For Authors:**

1.	How does the quality of the world model affect the final acceleration performance?
2.	Can SuP scale to real-world settings?

**Limitations:**

See weaknesses.

**Strengths And Weaknesses:**

**Strengths:**

1.	The proposed SuP is a plug-and-play policy acceleration method, which is scalable for diverse policies. More importantly, it does not require retraining the original policies.
2.	The optimization of the scheduler is formulated as a CMDP with a world model-based state deviation as the safety constraint. The world model can generate counterfactual trajectories by simulating various downsampling rates on offline data, which allows solving the CMDP in a pure offline setting.
3.	SuP effectively downsamples the action chunks and thus speeds up the policies.

**Weaknesses:**

1.	The final speedup effects of SuP may rely on the quality of the world model. A natural question is how to ensure the world model is reliable and how the quality of the world model affects the final acceleration performance.
2.	The proposed method was only evaluated on simulation tasks. Not sure if SuP can scale to real-world deployment and testing.
3.	The draft length is over 8 pages. The typesetting should be corrected. For example, the space between line 255 and line 256 is too small before Section 5.

---

> ### Author Rebuttal · Authors · 2026-03-31
>
> We highly appreciate your time and thoughtful remarks. We hereby clarify and respond to your critical concerns:
>
> >**W1\Q1:** How to ensure the world model is reliable and how the quality of the world model affects the final acceleration performance?
>
> We thank the reviewer for this important question. In our experiments, we trained the MAE loss of the world model to convergence with 3000 training steps. To study how world-model quality affects SuP, we evaluate checkpoints from different training stages and measure (i) prediction error (MAE), (ii) violation-success correlation (Sec. 5.4), and (iii) downstream SuP performance (success rate & steps). Results are summarized below.
>
> |Spatial|MAE|Correlation|Performance|
> |---|---:|---:|---|
> |ADM-Random|0.143|0.051|0.924,98.1|
> |ADM-Low|0.023|-0.158|0.956,86.2|
> |ADM-Middle|0.010|-0.492|0.970,73.6|
> |ADM-Final|0.008|-0.504|0.972,70.4|
>
> Two observations follow：
> 1. **Better world-model quality improves downstream acceleration:** from ADM-Random to ADM-Final, MAE drops markedly (0.143→0.008), the violation-performance correlation becomes much stronger (0.051→-0.504), and SuP achieves better downstream results. This suggests that a stronger world model provides more informative estimates of risky vs. safe acceleration choices.
> 2. **Once the world model is sufficiently good, SuP becomes less sensitive to further gains in raw prediction accuracy.** From ADM-Middle to ADM-Final, MAE still decreases (0.010→0.008), but the correlation changes little (-0.492 vs. -0.504), and downstream improvement is marginal. This demonstrates that the $\epsilon$ parameter acts as an effective buffer against prediction noise. As long as the world model's error is constrained within this $\epsilon$ margin, SuP can safely and consistently accelerate the downstream policy without requiring perfectly precise predictions.
>
> Overall, these results demonstrate that SuP is not overly demanding of perfectly high prediction accuracy. Instead, the clear trend between validation MAE and acceleration performance demonstrates that validation MAE is a reliable proxy for evaluating world-model quality. We will update the paper to explicitly discuss how to judge model reliability using this relationship.
>
> >**W2\Q2:** The proposed method was only evaluated on simulation tasks. Not sure if SuP can scale to real-world deployment and testing.
>
> We apologize that this was not sufficiently clear in the submission. In fact, Sec. 5.2 includes experiments on three real-robot tasks (arrange table, fold towel, and stack plates), where SuP consistently outperforms the baselines in both task success and efficiency. These results and properties suggest that SuP is applicable beyond simulation and can scale to real-world deployment.
>
> >**W3:** The draft length is over 8 pages. The typesetting should be corrected. For example, the space between line 255 and line 256 is too small before Section 5.
>
> We apologize for the formatting issues. The current draft slightly exceeds the page limit and contains suboptimal spacing in some parts (e.g., before Sec. 5). We will carefully revise the formatting and ensure that the final version strictly adheres to the page limits and improves readability.

---

> > ### Author Rebuttal · Reviewer_A78A · 2026-04-03
> >
> > Thanks for the additional results regarding world-model quality and clarifications. I will raise my score accordingly.

---

> > > ### Author Response · Authors · 2026-04-03
> > >
> > > We are happy that all your concerns have been addressed. We sincerely appreciate the reviewer for raising the score.

---

### Decision · Program_Chairs · 2026-04-30

**Decision:**

Accept (regular)

**Comment:**

This paper proposes SuP, a lightweight plug-and-play method to accelerate manipulation policies without retraining, formulated as a CMDP with world model-based safety constraints. All four reviewers raised their scores after the rebuttal, with all concerns fully resolved. The method is well-motivated, clearly presented, and validated across both simulation and real-world settings.